# PGDiff: Guiding Diffusion Models for Versatile Face Restoration via Partial Guidance

Peiqing Yang[1]    Shangchen Zhou[1]    Qingyi Tao[2]    Chen Change Loy[1]
[1]S-Lab, Nanyang Technological University
[2]SenseTime Research, Singapore
https://github.com/pq-yang/PGDiff

## Abstract

Exploiting pre-trained diffusion models for restoration has recently become a favored alternative to the traditional task-specific training approach. Previous works have achieved noteworthy success by limiting the solution space using explicit degradation models. However, these methods often fall short when faced with complex degradations as they generally cannot be precisely modeled. In this paper, we propose *PGDiff* by introducing *partial guidance*, a fresh perspective that is more adaptable to real-world degradations compared to existing works. Rather than specifically defining the degradation process, our approach models the desired properties, such as image structure and color statistics of high-quality images, and applies this guidance during the reverse diffusion process. These properties are readily available and make no assumptions about the degradation process. When combined with a diffusion prior, this partial guidance can deliver appealing results across a range of restoration tasks. Additionally, *PGDiff* can be extended to handle composite tasks by consolidating multiple high-quality image properties, achieved by integrating the guidance from respective tasks. Experimental results demonstrate that our method not only outperforms existing diffusion-prior-based approaches but also competes favorably with task-specific models.

## 1    Introduction

Recent years have seen diffusion models achieve outstanding results in synthesizing realistic details across various content [33, 29, 7, 15, 34]. The rich generative prior inherent in these models opens up a vast array of possibilities for tasks like super-resolution, inpainting, and colorization. Consequently, there has been a growing interest in formulating efficient guidance strategies for pre-trained diffusion models, enabling their successful adaptation to various restoration tasks [9, 42, 20, 37].

A common approach [9, 42, 20] is to constrain the solution space of intermediate outputs during the denoising process[1]. At each iteration, the intermediate output is modified such that its degraded counterpart is guided towards the input low-quality (LQ) image. Existing works achieve this goal either by using a closed-form solution [42, 20] or back-propagating simple losses [9]. These methods are versatile in the sense that the pre-trained diffusion model can be adapted to various tasks without fine-tuning, as long as the degradation process is known in advance.

While possessing great versatility, the aforementioned methods are inevitably limited in generalizability due to the need for prior knowledge of the degradation process. In particular, a closed-form solution generally does not exist except for special cases such as linear operators. In addition, back-propagating losses demand differentiability of the degradation process, which is violated for many degradations such as JPEG compression. Importantly, degradations in the wild often consist

---

[1]It refers to the reverse diffusion process, not the image denoising task.

37th Conference on Neural Information Processing Systems (NeurIPS 2023).

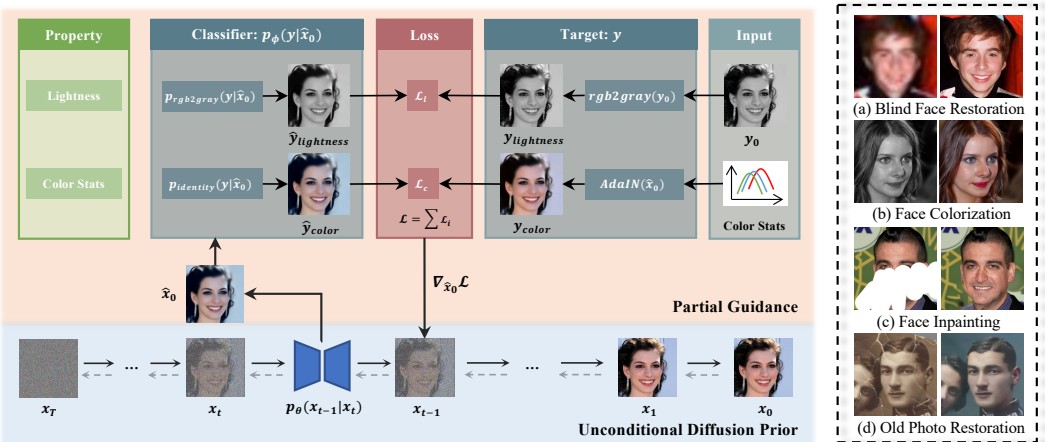

Figure 1: **Overview of Our PGDiff Framework for Versatile Face Restoration**. Here, we take the colorization task as an example to illustrate our inference pipeline. One may refer to Table 1 for the corresponding details (*e.g.*, property, classifier, and target) of other tasks. We show that our method can handle a wide range of tasks, including (a) blind face restoration, (b) face colorization, (c) face inpainting, and also composite tasks such as (d) old photo restoration.

of a mixture of degradations [41], and hence, it is difficult, if not impossible, to model them accurately. As a result, existing works generally limit the scope to simplified cases, such as fixed-kernel downsampling. The generalization to real-world degradations remains a formidable challenge.

Motivated by the above, instead of modeling the degradation process, we propose to model the *desired properties* of high-quality (HQ) images. The merit of such guidance is the agnosticity to the degradation process. However, it remains unclear what properties are desired and how appropriate guidance can be constructed. Through our extensive experiments, we find that with diffusion prior acting as a natural image regularization, one could simply guide the denoising process with easily accessible properties, such as image structure and color statistics. For example, as shown in Fig. 1, one could generate plausible outputs simply by providing guidance on the lightness and the statistics (*i.e.*, mean and variance) of each color channel, without knowing the exact decolorization process. By constraining the HQ image space, our idea bypasses the difficulty of knowing the prior relation between LQ and HQ images, thus improving generalizability.

In this work, we devise a simple yet effective instantiation named ***PGDiff*** by introducing ***partial guidace***. PGDiff adopts classifier guidance [7] to constrain the denoising process. Each image property corresponds to a classifier, and the intermediate outputs are updated by back-propagating the gradient computed on the loss between the classifier output and the target property. Since our partial guidance is agnostic to the degradation process, it can be easily extended to complex tasks by compositing multiple properties. For instance, the task of old photo restoration can be regarded as a combination of restoration, inpainting, and colorization, and the resultant guidance is represented as a weighted sum of the guidance in the respective task. We also demonstrate that common losses such as perceptual loss [2, 16] and adversarial loss [22] can be incorporated for further performance gain.

**Contributions.** Our main contributions include **i)** a new concept of adapting diffusion models to restoration without presumptions of the degradation process. We show that it suffices to guide the denoising process with *easily accessible properties* in the HQ image space, with diffusion prior acting as regularization, and **ii)** *partial guidance*, a versatile approach that is applicable to a broad range of image restoration and enhancement tasks. Furthermore, it allows flexible combinations of guidance for intricate tasks. We conduct extensive experiments to demonstrate the effectiveness of PGDiff on a variety of challenging tasks including blind face restoration and old photo restoration. We also demonstrate interesting applications, such as reference-based restoration. The results confirm the superiority of PGDiff over previous state-of-the-art methods.

## 2 Related Work

**Generative Prior for Restoration.** Generative prior has been widely adopted for a range of image restoration tasks, including super-resolution, inpainting, and colorization. One prominent approach in

this field is the use of pre-trained generative adversarial networks (GANs) [10, 18, 1]. For instance, GAN-inversion [26, 11, 28] inverts a corrupted image to a latent code, which is then used for generating a clean image. Another direction is to incorporate the prior into an encoder-decoder architecture [3, 4, 40, 44], bypassing the lengthy optimization during inference. VQVAE [35] is also commonly used as generative prior. Existing works [48, 13, 43, 47] generally first train a VQVAE with a reconstruction objective, followed by a fine-tuning stage to adapt to the subsequent restoration task. Recently, diffusion models have gained increasing attention due to their unprecedented performance in various generation tasks [33, 29, 7, 15, 34], and such attention has led to interest in leveraging them as a prior for restoration.

**Diffusion Prior.** There has been a growing interest in formulating efficient guidance strategies for pre-trained diffusion models, enabling their successful adaptation to various restoration tasks [9, 42, 20, 37, 39]. Among them, DDRM [20], DDNM [42], and GDP [9] adopt a zero-shot approach to adapt a pre-trained diffusion model for restoration without the need of task-specific training. At each iteration, the intermediate output is modified such that its degraded counterpart is guided towards the input low-quality image. This is achieved under an assumed degradation process, either in the form of a fixed linear matrix [42, 20] or a parameterized degradation model [9], with learnable parameters representing degradation extents. In this work, we also exploit the generative prior of a pre-trained diffusion model by formulating efficient guidance for it, but unlike existing works that limit the solution space using explicit degradations [9, 42, 20], we propose to model the desired properties of high-quality images. Such design is agnostic to the degradation process, circumventing the difficulty of modeling the degradation process.

## 3 Methodology

PGDiff is based on diffusion models. In this section, we first introduce the background related to our method in Sec. 3.1, and the details of our method are presented in Sec. 3.2.

### 3.1 Preliminary

**Diffusion Models.** The diffusion model [33] is a class of generative models that learn to model a data distribution $p(x)$. In particular, the forward process is a process that iteratively adds Gaussian noise to an input $x_0 \sim p(x)$, and the reverse process progressively converts the data from the noise distribution back to the data distribution, often known as the denoising process.

For an unconditional diffusion model with $T$ discrete steps, at each step $t$, there exists a transition distribution $q(x_t|x_{t-1})$ with variance schedule $\beta_t$ [15]:

$$q(x_t|x_{t-1}) = \mathcal{N}(x_t; \sqrt{1 - \beta_t}\, x_{t-1}, \beta_t \mathbf{I}). \tag{1}$$

Under the reparameterization trick, $x_t$ can be written as:

$$x_t = \sqrt{\alpha_t}\, x_{t-1} + \sqrt{1 - \alpha_t}\, \epsilon, \tag{2}$$

where $\alpha_t = 1 - \beta_t$ and $\epsilon \sim \mathcal{N}(\epsilon; \mathbf{0}, \mathbf{I})$. Recursively, let $\bar{\alpha}_t = \prod_{i=1}^{t} \alpha_i$, we have

$$x_t = \sqrt{\bar{\alpha}_t}\, x_0 + \sqrt{1 - \bar{\alpha}_t}\, \epsilon. \tag{3}$$

During sampling, the process starts with a pure Gaussian noise $x_T \sim \mathcal{N}(x_T; \mathbf{0}, \mathbf{I})$ and iteratively performs the denoising step. In practice, the ground-truth denoising step is approximated [7] by $p_\theta(x_{t-1}|x_t)$ as:

$$p_\theta(x_{t-1}|x_t) = \mathcal{N}(\mu_\theta(x_t, t), \Sigma_\theta(x_t, t)), \tag{4}$$

where $\Sigma_\theta(x_t, t)$ is a constant depending on pre-defined $\beta_t$, and $\mu_\theta(x_t, t)$ is generally parameterized by a network $\epsilon_\theta(x_t, t)$:

$$\mu_\theta(x_t, t) = \frac{1}{\sqrt{\alpha_t}}(x_t - \frac{\beta_t}{\sqrt{1 - \bar{\alpha}_t}}\epsilon_\theta(x_t, t)). \tag{5}$$

From Eq. (3), one can also directly approximate $x_0$ from $\epsilon_\theta$:

$$\hat{x}_0 = \frac{1}{\sqrt{\bar{\alpha}_t}}x_t - \sqrt{\frac{1 - \bar{\alpha}_t}{\bar{\alpha}_t}}\epsilon_\theta(x_t, t). \tag{6}$$

**Algorithm 1** Sampling with *partial guidance*, given a diffusion model $(\mu_\theta(x_t, t), \Sigma_\theta(x_t, t))$, classifier $p_\phi(y|\hat{x}_0)$, target $y$, gradient scale $s$, range for multiple gradient steps $S_{start}$ and $S_{end}$, and the number of gradient steps $N$. The dynamic guidance weight $s_{norm}$ is omitted for brevity.

---

**Input**: a low-quality image $y_0$
$x_T \leftarrow$ sample from $\mathcal{N}(\mathbf{0}, \mathbf{I})$
**for** $t$ from $T$ to 1 **do**
    $\mu, \Sigma \leftarrow \mu_\theta(x_t, t), \Sigma_\theta(x_t, t)$
    $\hat{x}_0 \leftarrow \frac{1}{\sqrt{\bar{\alpha}_t}} x_t - \sqrt{\frac{1-\bar{\alpha}_t}{\bar{\alpha}_t}} \epsilon_\theta(x_t, t)$
    **if** $S_{start} \le t \le S_{end}$ **then**                           ▷ Multiple Gradient Steps
        **repeat**
            $x_t \leftarrow$ sample from $\mathcal{N}(\mu - s\Sigma\nabla_{\hat{x}_0}\|y - p_\phi(y|\hat{x}_0)\|_2^2, \Sigma)$
            $\hat{x}_0 \leftarrow \frac{1}{\sqrt{\bar{\alpha}_t}} x_t - \sqrt{\frac{1-\bar{\alpha}_t}{\bar{\alpha}_t}} \epsilon_\theta(x_t, t)$
        **until** $N - 1$ times
    **end if**
    $x_{t-1} \leftarrow$ sample from $\mathcal{N}(\mu - s\Sigma\nabla_{\hat{x}_0}\|y - p_\phi(y|\hat{x}_0)\|_2^2, \Sigma)$
**end for**
**return** $x_0$

---

Table 1: **Examples of Partial Guidance.** Each image property corresponds to a classifier, and each task involves one or multiple properties as guidance. The target value of each property is generally obtained either from the input image $y_0$ or the denoised intermediate output $\hat{x}_0$. For a composite task, we simply decompose it into multiple tasks and combine the respective guidance. Here, `Clean` denotes a pre-trained restorer detailed in Sec. 4.1, `Identity` refers to an identity mapping, and $y_{ref}$ represents a reference image containing entity with the same identity as $y_0$.

| | Task | Property | Target: $y$ | Classifier: $p_\phi(y|\hat{x}_0)$ |
|---|---|---|---|---|
| **Homogeneous Task** | Inpainting | Unmasked Region | `Mask`$(y_0)$ | `Mask` |
| | Colorization | Lightness | `rgb2gray`$(y_0)$ | `rgb2gray` |
| | | Color Statistics | `AdaIN`$(\hat{x}_0)$ [18] | `Identity` |
| | Restoration | Smooth Semantics | `Clean`$(y_0)$ | `Identity` |
| | Ref-Based Restoration | Smooth Semantics | `Clean`$(y_0)$ | `Identity` |
| | | Identity Reference | `ArcFace`$(y_{ref})$ | `ArcFace` [6] |
| | **Task** | **Composition** | | |
| **Composite Task** | Old Photo Restoration (w/ scratches) | Restoration + Inpainting + Colorization | | |

**Classifier Guidance.** Classifier guidance is used to guide an unconditional diffusion model so that conditional generation is achieved. Let $y$ be the target and $p_\phi(y|x)$ be a classifier, the conditional distribution is approximated as a Gaussian similar to the unconditional counterpart, but with the mean shifted by $\Sigma_\theta(x_t, t)g$ [7]:

$$p_{\theta,\phi}(x_{t-1}|x_t, y) \approx \mathcal{N}(\mu_\theta(x_t, t) + \Sigma_\theta(x_t, t)g, \Sigma_\theta(x_t, t)), \tag{7}$$

where $g = \nabla_x \log p_\phi(y|x)|_{x=\mu_\theta(x_t,t)}$. The gradient $g$ acts as a guidance that leads the unconditional sampling distribution towards the condition target $y$.

### 3.2 Partial Guidance

Our *partial guidance* does not assume any prior knowledge of the degradation process. Instead, with diffusion prior acting as a regularization, we provide guidance only on the desired properties of high-quality images. The key to PGDiff is to construct proper guidance for each task. In this section, we will discuss the overall framework, and the formulation of the guidance for each task is presented in Sec. 4. The overview is summarized in Fig. 1 and Algorithm 1.

**Property and Classifier.** The first step of PGDiff is to determine the desired properties which the high-quality output possesses. As summarized in Table 1, each image property corresponds to a classifier $p_\phi(y|\hat{x}_0)$, and the intermediate outputs $x_t$ are updated by back-propagating the gradient computed on the loss between the classifier output and the target $y$.

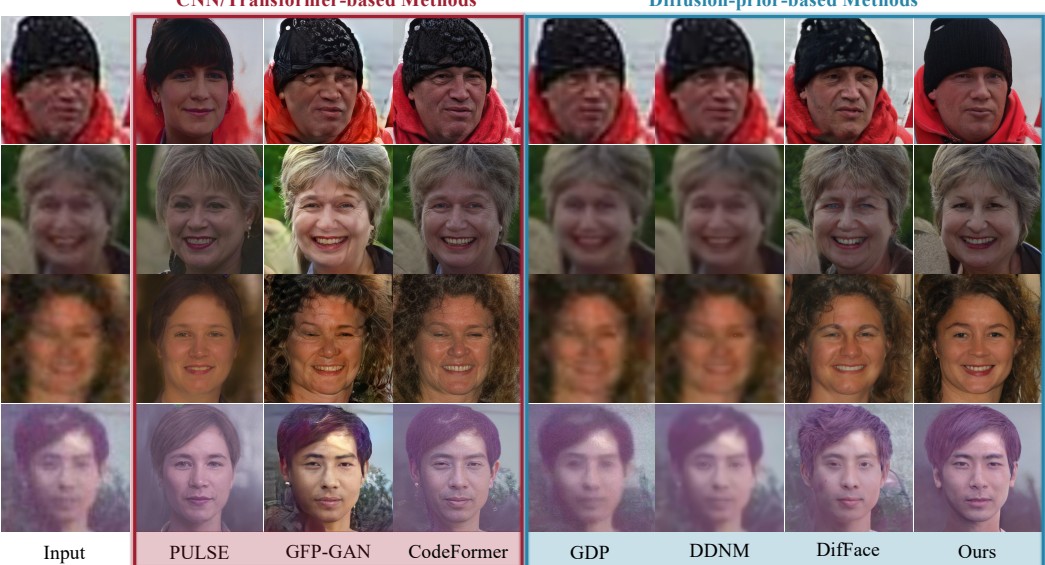

Figure 2: **Comparison on Blind Face Restoration.** Input faces are corrupted by real-world degradations. Our PGDiff produces high-quality faces with faithful details. (**Zoom in for best view.**)

Given a specific property (*e.g.*, lightness), we construct the corresponding classifier (*e.g.*, `rgb2gray`), and apply classifier guidance during the reverse diffusion process as shown in Fig. 1. Although our PGDiff is conceptually similar to classifier guidance, we find that the conventional guidance scheme often leads to suboptimal performance. In this work, we borrow ideas from existing works [37, 5] and adopt a *dynamic guidance scheme*, which introduces adjustments to the guidance weight and number of gradient steps for enhanced quality and controllability.

**Dynamic Guidance Scheme.** Our dynamic guidance scheme consists of two components. First, we observe that the conventional classifier guidance, which adopts a constant gradient scale $s$, often fails in guiding the output towards the target value. This is especially unfavourable in tasks where high similarity to the target is desired, such as inpainting and colorization. To alleviate this problem, we calculate the gradient scale based on the magnitude change of the intermediate image [37]:

$$s_{norm} = \frac{\|x_t - x'_{t-1}\|_2}{\|g\|_2} \cdot s,$$

(8)

where $x'_{t-1} \sim \mathcal{N}(\mu_\theta, \Sigma_\theta)$. In this way, the dynamic guidance weight $s_{norm}$ varies along iterations, more effectively guiding the output towards the target, thus improving the output quality.

Second, the conventional classifier guidance typically executes a single gradient step at each denoising step. However, a single gradient step may not sufficiently steer the output toward the intended target, particularly when the intermediate outputs are laden with noise in the early phases of the denoising process. To address this, we allow multiple gradient steps at each denoising step [5] to improve flexibility. Specifically, one can improve the guidance strength of a specific property by increasing the number of gradient steps. The process degenerates to the conventional classifier guidance when the number of gradient steps is set to 1. During inference, users have the flexibility to modulate the strength of guidance for each property as per their requirements, thus boosting overall controllability.

**Composite Guidance.** Our partial guidance controls only the properties of high-quality outputs, and therefore can be easily extended to complex degradations by stacking respective properties. This is achieved by compositing the classifiers and summing the loss corresponding to each property. An example of composite tasks is shown in Table 1. In addition, we also demonstrate that additional losses such as perceptual loss [2, 16] and adversarial loss [22] can be incorporated for further quality improvement. Experiments demonstrate that our PGDiff achieves better performance than existing works in complex tasks, where accurate modeling of the degradation process is impossible.

Table 2: Quantitative comparison on the *real-world* **LFW-Test**, **WebPhoto-Test**, and **WIDER-Test** datasets. Red and blue indicate the best and the second best performance, respectively.

| Dataset | Metric | CNN/Transformer-based Methods | | Diffusion-prior-based Methods | |
|---|---|---|---|---|---|
| | | GFP-GAN [40] | CodeFormer [48] | DifFace [45] | Ours |
| **LFW-Test** | FID↓ | 72.45 | 74.10 | 67.98 | 71.62 |
| | NIQE↓ | 3.90 | 4.52 | 5.47 | 4.15 |
| **WebPhoto-Test** | FID↓ | 91.43 | 86.19 | 90.58 | 86.18 |
| | NIQE↓ | 4.13 | 4.65 | 4.48 | 4.34 |
| **WIDER-Test** | FID↓ | 40.93 | 40.26 | 38.54 | 39.17 |
| | NIQE↓ | 3.77 | 4.12 | 4.44 | 3.93 |

Table 3: Quantitative comparison on the *synthetic* **CelebRef-HQ** dataset. Red and blue indicate the best and the second best performance, respectively.

| Metric | CNN/Transformer-based Methods | | Diffusion-prior-based Methods | | |
|---|---|---|---|---|---|
| | GFP-GAN [40] | CodeFormer [48] | DifFace [45] | Ours (w/o ref) | Ours (w/ ref) |
| FID↓ | 186.88 | 129.17 | 123.18 | 119.98 | 121.25 |
| MUSIQ↑ | 63.33 | 69.62 | 60.98 | 67.26 | 64.67 |
| LPIPS↓ | 0.49 | 0.36 | 0.35 | 0.34 | 0.35 |
| IDS↑ | 0.36 | 0.55 | 0.56 | 0.44 | 0.76 |

# 4 Applications

By exploiting the diffusion prior, our PGDiff applies to a wide range of restoration tasks by selecting appropriate guidance. In this section, we will introduce the guidance formulation and provide experimental results.

## 4.1 Blind Face Restoration

**Partial Guidance Formulation.** The objective of blind face restoration is to reconstruct a high-quality face image given a low-quality input corrupted by unknown degradations. In this task, the most straightforward approach is to train a network with the MSE loss using synthetic pairs. However, while these methods are able to remove the degradations in the input, it is well-known [26] that the MSE loss alone results in over-smoothed outputs. Therefore, extensive efforts have been devoted to improving the perceptual quality, such as incorporating addition losses (*e.g.*, GAN loss) [22, 10, 16, 46, 8] and components (*e.g.*, codebook [48, 13, 43, 47, 35] and dictionary [23, 24, 12, 8]). These approaches often require multi-stage training and experience training instability.

In our framework, we decompose a high-quality face image into *smooth semantics* and *high-frequency details*, and provide guidance solely on the *smooth semantics*. In this way, the output $\hat{x}_0$ in each diffusion step is guided towards a degradation-free solution space, and the diffusion prior is responsible for detail synthesis. Given an input low-quality image $y_0$, we adopt a pre-trained face restoration model $f$ to predict smooth semantics as partial guidance. Our approach alleviates the training pressure of the previous models by optimizing model $f$ solely with the MSE loss. This is because our goal is to obtain *smooth semantics* without hallucinating unnecessary high-frequency details. Nevertheless, one can also provide guidance of various forms by selecting different restorers, such as CodeFormer [48]. The loss for classifier guidance is computed as: $\mathcal{L}_{res} = ||\hat{x}_0 - f(y_0)||_2^2$.

**Qualitative Results.** We evaluate the proposed PGDiff on three real-world datasets, namely LFW-Test [40], WebPhoto-Test [40], and WIDER-Test [48]. We compare our method with both task-specific CNN/Transformer-based restoration models [48, 26, 40] and diffusion-prior-based models[2] [9, 42, 45]. As shown in Fig. 2, existing diffusion-prior-based methods such as GDP [9] and DDNM [42] are unable to generalize to real-world degradations, producing outputs with notable artifacts. In contrast, our PGDiff successfully removes the degradations and restores the facial details invisible in the input

---

[2]Among them, GDP [9] and DDNM [42] support only $4\times$ fixed-kernel downsampling, while DifFace [45] is a task-specific model for blind face restoration.

images. Moreover, our PGDiff performs favorably over task-specific methods even without extensive training on this task.

**Quantitative Results on Real-world Datasets.** To compare our performance with other methods quantitatively on real-world datasets, we adopt FID [14] and NIQE [27] as the evaluation metrics and test on three real-world datasets: LFW-Test [40], WebPhoto-Test [40], and WIDER-Test [48]. LFW-Test consists of the first image from each person whose name starts with `A` in the LFW dataset [40], which are 431 images in total. WebPhoto-Test is a dataset comprising 407 images with medium degradations collected from the Internet. WIDER-Test contains 970 severely degraded images from the WIDER Face dataset [40]. As shown in Table 2, our method achieves the best or second-best scores across all three datasets for both metrics. Although GFP-GAN achieves the best NIQE scores across datasets, notable artifacts can be observed, as shown in Fig. 2. Meanwhile, our method shows exceptional robustness and produces visually pleasing outputs without artifacts.

**Quantitative Results on Synthetic Dataset.** We present a quantitative evaluation on the synthetic CelebRef-HQ dataset [24] in Table 3. Considering the importance of identity-preserving in blind face restoration, we introduce reference-based restoration in Sec. 4.5 in addition to the general restoration in Sec. 4.1. Table 3 shows that our methods achieve best or second best scores across both no-reference (NR) metrics for image quality (*i.e.*, FID and MUSIQ) and full-reference (FR) metrics for identity preservation (*i.e.*, LPIPS and IDS). Since we employ heavy degradation settings when synthesizing CelebRef-HQ, it is noteworthy that identity features are largely distorted in severely corrupted input images. Thus, it is almost impossible to predict an identity-preserving face without any additional identity information. Nevertheless, with our reference-based restoration, we observe that a high-quality reference image of the same person helps generate personal characteristics that are highly similar to the ground truth. The large enhancement of identity preservation is also indicated in Table 3, where our reference-based method achieves the highest IDS, increasing by 0.32.

## 4.2 Face Colorization

**Partial Guidance Formulation.** Motivated by color space decomposition (*e.g.*, YCbCr, YUV), we decompose our guidance into *lightness* and *color*, and provide respective guidance on the two aspects. For lightness, the input image acts as a natural target since it is a homogeneous-color image. Specifically, we guide the output lightness towards that of the input using the simple `rgb2gray` operation. Equivalently, the loss is formulated as follows: $\mathcal{L}_l = ||\texttt{rgb2gray}(\hat{x}_0) - \texttt{rgb2gray}(y_0)||_2^2$. The lightness guidance can also be regarded as a dense structure guidance. This is essential in preserving image content.

With the lightness guidance constraining the structure of the output, we could guide the color synthesis process with a lenient constraint – color statistics (*i.e.*, mean and variance of each color channel). In particular, we construct the target by applying `AdaIN` [18] to $\hat{x}_0$, using a pre-determined set of color statistics for each R, G, B channel. Then we push $\hat{x}_0$ towards the color-normalized output: $\mathcal{L}_c = ||\hat{x}_0 - \texttt{sg}\left(\texttt{AdaIN}(\hat{x}_0, \mathbb{P})\right)||_2^2$, where $\mathbb{P}$ refers to the set of color statistics and $\texttt{sg}(\cdot)$ denotes the stop-gradient operation [35]. The overall loss is formulated as: $\mathcal{L}_{color} = \mathcal{L}_l + \alpha \cdot \mathcal{L}_c$, where $\alpha$ is a constant that controls the relative importance of the structure and color guidance. To construct a universal color tone, we compute the average color statistics from a selected subset of the CelebA-HQ dataset [17]. We find that this simple strategy suffices to produce faithful results. Furthermore, our PGDiff can produce outputs with diverse color styles by computing the color statistics from different reference images.

**Experimental Results.** As shown in Fig. 3, GDP [9] and DDNM [42] lack the capability to produce vibrant colors. In contrast, our PGDiff produces colorized outputs simply by modeling the lightness and color statistics. Furthermore, our method is able to generate outputs with diverse color styles by calculating color statistics from various reference sets.

## 4.3 Face Inpainting

**Partial Guidance Formulation.** Since diffusion models have demonstrated remarkable capability in synthesizing realistic content [33, 29, 7, 15, 34], we apply guidance only on the unmasked regions, and rely on the synthesizing power of diffusion models to generate details in the masked regions. Let $B$ be a binary mask where $0$ and $1$ denote the masked and unmasked regions, respectively. We confine the solution by ensuring that the resulting image closely resembles the input image

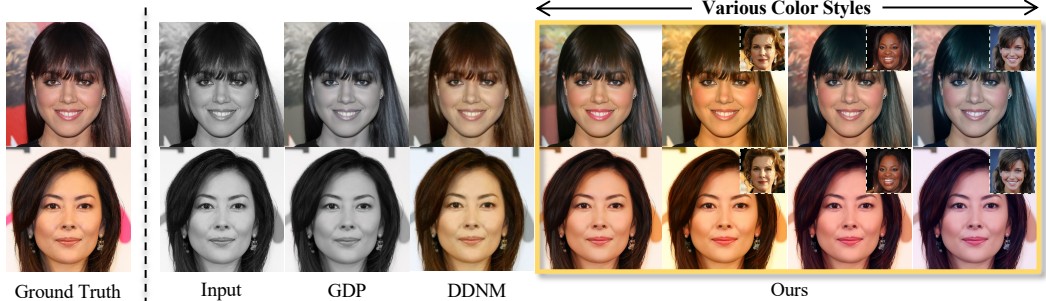

Figure 3: **Comparison on Face Colorization.** Our PGDiff produces diverse colorized output with various color statistics given as guidance. The first column of our results is guided by the average color statistics of a subset of the CelebA-HQ dataset [17], and the guiding statistics for the remaining three columns are represented as an image in the top right corner.

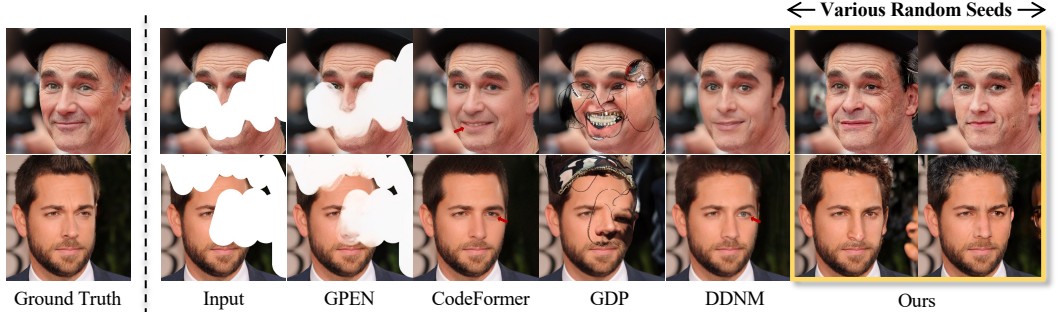

Figure 4: **Comparison on Face Inpainting on Challenging Cases.** Our PGDiff produces natural outputs with pleasant details coherent with the unmasked regions. Moreover, different random seeds give various contents of high quality.

within the unmasked regions: $\mathcal{L}_{inpaint} = ||B \otimes \hat{x}_0 - B \otimes y_0||_2^2$, where $\otimes$ represents the pixel-wise multiplication.

**Experimental Results.** We conduct experiments on CelebRef-HQ [24]. As depicted in Fig. 4, GPEN [44] and GDP [9] are unable to produce natural outputs, whereas CodeFormer [48] and DDNM [42] generate outputs with artifacts, such as color incoherence or visual flaws. In contrast, our PGDiff successfully generates outputs with pleasant details coherent to the unmasked regions.

### 4.4 Old Photo Restoration

**Partial Guidance Formulation.** Quality degradations (*e.g.*, blur, noise, downsampling, and JPEG compression), color homogeneity, and scratches are three commonly seen artifacts in old photos. Therefore, we cast this problem as a joint task of *restoration*, *colorization*, and *inpainting*[3]. Similar to face colorization that composites the loss for each property, we composite the respective loss in each task, and the overall loss is written as: $\mathcal{L}_{old} = \mathcal{L}_{res} + \gamma_{color} \cdot \mathcal{L}_{color} + \gamma_{inpaint} \cdot \mathcal{L}_{inpaint}$, where $\gamma_{inpaint}$ and $\gamma_{color}$ are constants controlling the relative importance of the different losses.

**Experimental Results.** We compare our PGDiff with BOPB [38], GFP-GAN [40] and DDNM [42]. Among them, BOPB is a model specifically for old photo restoration, GFP-GAN (v1) is able to restore and colorize faces, and DDNM is a diffusion-prior-based method that also claims to restore old photos with scratches. As shown in Fig. 5, BOPB, GFP-GAN, and DDNM all fail to give natural color in such a composite task. While DDNM is able to complete scratches given a proper scratch map, it fails to give a high-quality face restoration result. On the contrary, PGDiff generates sharp colorized faces without scratches and artifacts.

---

[3]We locate the scratches using an automated algorithm [38], and then inpaint the scratched regions.

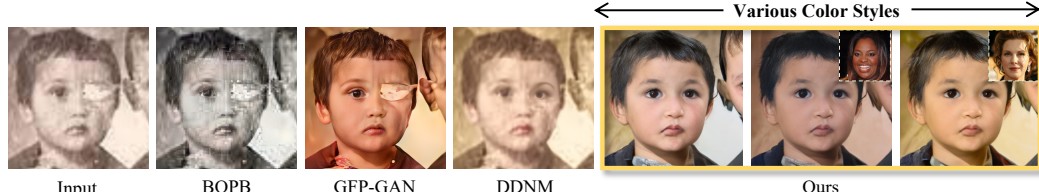

Input      BOPB      GFP-GAN      DDNM         Ours

Figure 5: **Comparison on Old Photo Restoration on Challenging Cases.** For a severely damaged old photo, with one eye masked with scratch, while only DDNM [42] is able to complete the missing eye, its restoration quality is significantly low. In contrast, our PGDiff produces high-quality restored outputs with natural color and complete faces.

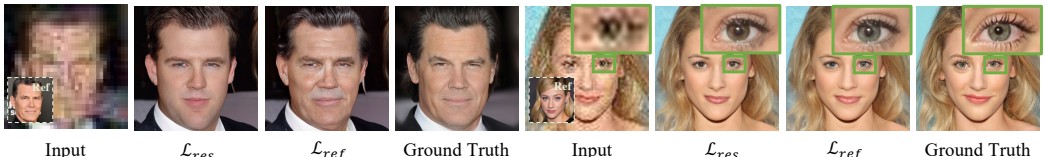

Input    $\mathcal{L}_{res}$    $\mathcal{L}_{ref}$    Ground Truth    Input    $\mathcal{L}_{res}$    $\mathcal{L}_{ref}$    Ground Truth

Figure 6: **Reference-Based Face Restoration.** Our PGDiff, using $\mathcal{L}_{ref}$ with identity loss as guidance, produces personal characteristics that are hard to recover without reference, *i.e.*, using $\mathcal{L}_{res}$ only. **(Zoom in for details)**

## 4.5 Reference-Based Restoration

**Partial Guidance Formulation.** In reference-based restoration, a reference image from the same identity is given to improve the resemblance of personal details in the output image. Most existing works exploiting diffusion prior [9, 42, 20, 45] are not applicable to this task as there is no direct transformation between the reference and the target. In contrast, our partial guidance is extensible to more complex tasks simply by compositing multiple losses. In particular, our PGDiff can incorporate personal identity as a partial attribute of a facial image. By utilizing a reference image and incorporating the identity loss into the partial guidance, our framework can achieve improved personal details. We extract the identity features from the reference image using a pre-trained face recognition network, such as ArcFace [6]. We then include the negative cosine similarity to the loss term $\mathcal{L}_{res}$ in blind face restoration (Sec. 4.1): $\mathcal{L}_{ref} = \mathcal{L}_{res} - \beta \cdot \text{sim}(v_{\hat{x}_0}, v_r)$, where $\beta$ controls the relative weight of the two losses. Here $\text{sim}(\cdot)$ represents the cosine similarity, and $v_{\hat{x}_0}$ and $v_r$ denote the ArcFace features of the predicted denoised image and the reference, respectively.

**Experimental Results.** We use the CelebRef-HQ dataset [24], which contains $1,005$ entities and each person has 3 to 21 high-quality images. To build testing pairs, for each entity, we choose one image and apply heavy degradations as the input, and then we select another image from the same identity as the reference. In Fig. 6, we observe that without the identity loss term $\text{sim}(v_{\hat{x}_0}, v_r)$, some of the personal details such as facial wrinkles and eye color cannot be recovered from the distorted inputs. With the additional identity loss as guidance, such fine details can be restored. In addition, our PGDiff can be used to improve identity preservation of arbitrary face restorers. For instance, as shown in Fig. 7 (a), by using CodeFormer [48] as our restorer and incorporating the identity loss, the fine details that CodeFormer alone cannot restore can now be recovered. Quantitative results are included in Table 3 and discussed in Sec. 4.1.

## 4.6 Quality Enhancement

**Partial Guidance Formulation.** Perceptual loss [16] and adversarial loss [10] are two common training losses used to improve quality. Motivated by this, we are interested in whether such losses can also be used as the guidance for additional quality gain. We demonstrate this possibility in the task of blind face restoration using the following loss: $\mathcal{L}_{quality} = \mathcal{L}_{res} + \lambda_{per} \cdot ||\text{VGG}(\hat{x}_0) - \text{VGG}(y)||_2^2 + \lambda_{GAN} \cdot D(\hat{x}_0)$, where $\lambda_{per}$ and $\lambda_{GAN}$ are the relative weights. Here VGG and $D$ represent pre-trained VGG16 [32] and the GAN discriminator [19], respectively.

**Experimental Results.** We demonstrate in Fig. 7 (b) that perceptual loss and adversarial loss can boost the blind restoration performance in terms of higher fidelity with photo-realism details.

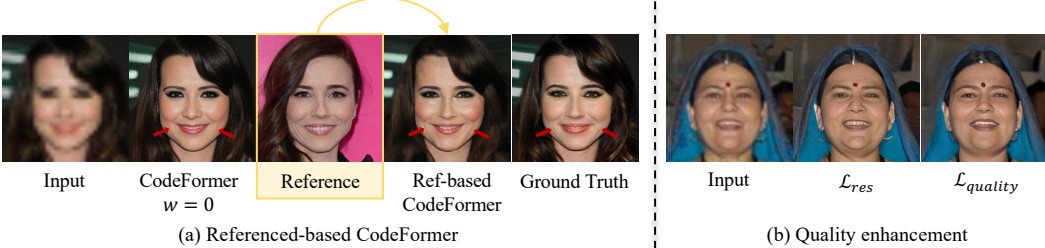

| Input | CodeFormer $w = 0$ | Reference | Ref-based CodeFormer | Ground Truth | | Input | $\mathcal{L}_{res}$ | $\mathcal{L}_{quality}$ |

(a) Referenced-based CodeFormer (b) Quality enhancement

Figure 7: (a) Using CodeFormer as the restorer with our identity guidance improves the reconstruction of fine details similar to the ground truth. (b) The comparison results show that the quality enhancement loss is able to enhance fidelity with photo-realism details.

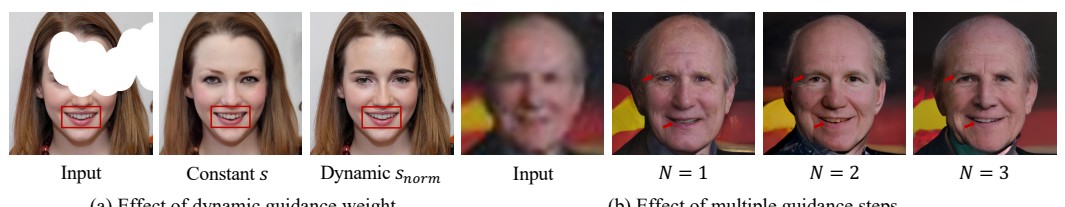

| Input | Constant $s$ | Dynamic $s_{norm}$ | Input | $N = 1$ | $N = 2$ | $N = 3$ |

(a) Effect of dynamic guidance weight (b) Effect of multiple guidance steps

Figure 8: **Ablation Study of Dynamic Guidance.** The comparison results on the dynamic guidance scheme verify its effectiveness over the conventional classifier guidance scheme.

## 5 Ablation Studies

In this section, we perform ablation studies on the dynamic guidance scheme mentioned in Sec. 3.2 to verify its effectiveness over the conventional classifier guidance scheme.

**Effectiveness of Dynamic Guidance Weight.** We first investigate the effectiveness of dynamic guidance weight $s_{norm}$ in the face inpainting task, where the unmasked regions of the output image should be of high similarity to that of the input. As shown in Fig. 8 (a), without the dynamic guidance weight, although plausible content can still be generated in the masked area, the similarity and sharpness of the unmasked regions are remarkably decreased compared with the input. With $s_{norm}$ replacing the constant $s$, the output is of high quality with unmasked regions nicely preserved. The results indicate that our dynamic guidance weight is the key to ensuring high similarity to the target during the guidance process.

**Effectiveness of Multiple Gradient Steps.** To verify the effectiveness of multiple gradient steps, we compare the blind restoration results with the number of guidance steps $N$ set to be 1, 2, and 3. While $N = 1$ is just the conventional classifier guidance, we set $N = 2$ during the first $0.5T$ steps and set $N = 3$ during the first $0.3T$ steps. As shown in Fig. 8 (b), artifacts are removed and finer details are generated as $N$ increases. These results suggest that multiple gradient steps serve to improve the strength of guiding the output toward the intended target, particularly when the intermediate outputs are laden with noise in the early phases of the denoising process.

## 6 Conclusion

The generalizability of existing diffusion-prior-based restoration approaches is limited by their reliance on prior knowledge of the degradation process. This study aims to offer a solution that alleviates this constraint, thereby broadening its applicability to the real-world degradations. We find that through directly modeling high-quality image properties, one can reconstruct faithful outputs without knowing the exact degradation process. We exploit the synthesizing power of diffusion models and provide guidance only on properties that are easily accessible. Our proposed *PGDiff* with *partial guidance* is not only effective but is also extensible to composite tasks through aggregating multiple properties. Experiments demonstrate that PGDiff outperforms diffusion-prior-based approaches in both homogeneous and composite tasks and matches the performance of task-specific methods.

**Acknowledgement.** This study is supported under the RIE2020 Industry Alignment Fund – Industry Collaboration Projects (IAF-ICP) Funding Initiative, as well as cash and in-kind contribution from the industry partner(s).

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

# Appendix

In this supplementary material, we provide additional discussions and results. In Sec. A, we present additional implementation details including the inference requirements, choice of hyperparameters involved in the inference process, and discussions on the pre-trained restorer for blind face restoration. In Sec. B, we provide more results on various tasks, *i.e.*, blind face restoration, old photo restoration, reference-based restoration, face colorization, and inpainting. Sec. C and Sec. D discuss the limitations and potential negative societal impacts of our work, respectively.

## A  Implementation Details

### A.1  Inference Requirements

The pre-trained diffusion model we employ is a $512 \times 512$ denoising network trained on the FFHQ dataset [18] provided by [45]. The inference process is carried out on NVIDIA RTX A5000 GPU.

### A.2  Inference Hyperparameters

During the inference process, there involves hyperparameters belonging to three categories. **(1) Sampling Parameters:** The parameters in the sampling process (*e.g.*, gradient scale $s$). **(2) Partial Guidance Parameters:** Additional parameters introduced by our partial guidance, which are mainly relative weights for properties involved in a certain task (*e.g.*, $\alpha$ that controls the relative importance between the structure and color guidance in face colorization). **(3) Optional Parameters:** Parameters for optional quality enhancement (*e.g.*, the range for multiple gradient steps to take place $[S_{start}, S_{end}]$). While it is principally flexible to tune the hyperparameters case by case, we provide a set of default parameter choices for each homogeneous task in Table 4.

Table 4: Default hyperparameter settings in our experiments.

| Task | Sampling | | Partial Guidance | | | | | Optional | | | |
|---|---|---|---|---|---|---|---|---|---|---|---|
| | $s_{norm}$ | $s$ | Unmasked Region | Lightness | Color Statistics | Smooth Semantics | Identity Reference | $N=2$ | $N=3$ | Perceptual Loss | GAN Loss |
| Restoration | | 0.1 | - | - | - | $\mathcal{L}_{res}$ | - | $T \sim 0.5T$ | $T \sim 0.7T$ | 1e-2 | 1e-2 |
| Colorization | ✓ | 0.01 | - | $\mathcal{L}_l$ | $0.01\mathcal{L}_c$ | - | - | - | - | - | - |
| Inpainting | ✓ | 0.01 | $\mathcal{L}_{inpaint}$ | - | - | - | - | - | - | - | - |
| Ref-Based Restoration | | 0.1 | - | - | - | $\mathcal{L}_{res}$ | $10\,\texttt{sim}(v_{\hat{x}_0}, v_r)$ | $T \sim 0.5T$ | $T \sim 0.7T$ | 1e-2 | 1e-2 |

### A.3  Restorer Design

**Network Structure.** In the blind face restoration task, given an input low-quality (LQ) image $y_0$, we adopt a pre-trained face restoration model $f$ to predict smooth semantics as partial guidance. In this work, we employ the $\times 1$ generator of Real-ESRGAN [41] as our restoration backbone. The network follows the basic structure of SRResNet [22], with RRDB being its basic blocks. In a $\times 1$ generator, the input image is first downsampled $4$ times by a pixel unshuffling [31] layer before any convolution operations. In our work, we deal with $512 \times 512$ input/output pairs, which means that most computation is done only in a $128 \times 128$ resolution scale. To employ it as the restorer $f$, we modify some of its settings. Empirically we find that adding $x_t$ and $t$ as the input alongside $y_0$ can enhance the sample quality in terms of sharpness. Consequently, the input to $f$ is a concatenation of $y_0$, $x_t$, and $t$, with $t$ embedded with the sinusoidal timestep embeddings [36].

**Training Details.** $f$ is implemented with the PyTorch framework and trained using four NVIDIA Tesla V100 GPUs at 200K iterations. We train $f$ with the FFHQ [18] and CelebA-HQ [17] datasets and form training pairs by synthesizing LQ images $I_l$ from their HQ counterparts $I_h$, following a common pipeline with a second-order degradation model [41, 23, 40, 44]. Since our goal is to obtain *smooth semantics* without hallucinating unnecessary high-frequency details, **it is sufficient to optimize the model $f$ solely with the MSE loss**.

**Model Analysis.** To investigate the most effective restorer for blind face restoration, we compare the sample quality with restorer being $f(y_0)$ and $f(y_0, x_t, t)$, respectively. Here, $f(y_0, x_t, t)$ is the one trained by ourselves as discussed above, and $f(y_0)$ is SwinIR [25] from DifFace [45], which is also

Table 5: Quantitative comparison on the *synthetic* **CelebA-Test** dataset on inpainting and colorization tasks. Red indicates the best performance.

| Task | Inpainting | | | Colorization | | |
|---|---|---|---|---|---|---|
| Metric | FID↓ | NIQE↓ | MUSIQ-KonIQ↑ | FID↓ | NIQE↓ | MUSIQ-AVA↑ |
| DDNM [42] | 137.57 | 5.35 | 59.38 | 146.66 | 5.11 | 4.07 |
| CodeFormer [48] | 120.93 | 4.22 | 72.48 | 126.91 | 4.43 | 4.91 |
| **Ours** | 115.99 | 3.65 | 73.20 | 119.31 | 4.71 | 5.23 |

trained with MSE loss only. As shown in Fig. 9, when all the other inference settings are the same, we find that the sample quality with restorer $f(y_0, x_t, t)$ is higher in terms of sharpness compared with that of $f(y_0)$. One may sacrifice a certain degree of sharpness to achieve higher inference speed by substituting the restorer with $f(y_0)$, whose output is constant throughout $T$ timesteps.

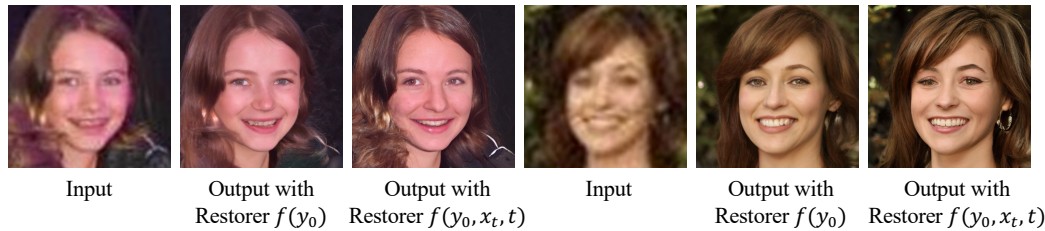

| Input | Output with Restorer $f(y_0)$ | Output with Restorer $f(y_0, x_t, t)$ | Input | Output with Restorer $f(y_0)$ | Output with Restorer $f(y_0, x_t, t)$ |

Figure 9: Visual comparison of the restoration outputs with different restorers $f$ in blind restoration. We observe that including $x_t$ and $t$ as the input to $f$ enhances the sharpness of the restored images.

# B More Results

## B.1 More Results on Blind Face Restoration

In this section, we provide more qualitative comparisons with state-of-the-art methods, including (1) task-specific CNN/Transformer-based restoration methods: PULSE [26], GFP-GAN [40], and CodeFormer [48] and (2) diffusion-prior-based methods: GDP [9], DDNM [42] and DifFace [45].

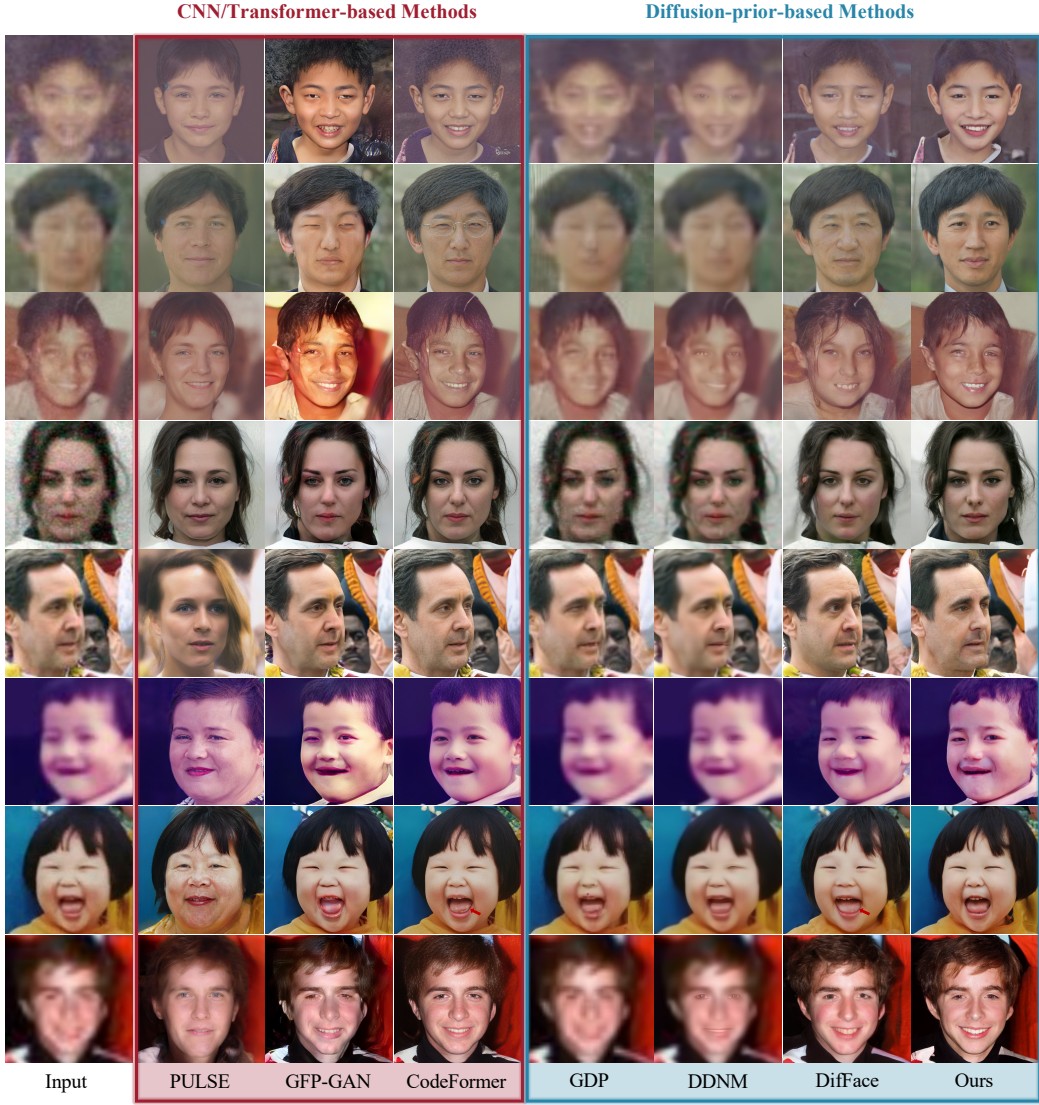

Figure 10: **Comparison on Blind Face Restoration.** Input faces are corrupted by real-world degradations. Our PGDiff produces high-quality faces with faithful details. (**Zoom in for best view**)

## B.2  More Results on Old Photo Restoration

We provide more visual results of old photo restoration on challenging cases both with and without scratches, as shown in Fig. 11. The test images come from both the CelebChild-Test dataset [40] and the Internet. We compare our method with GFP-GAN (v1) [40] and DDNM [42]. Our method demonstrates an obvious advantage in sample quality, especially in terms of vibrant colors, fine details, and sharpness.

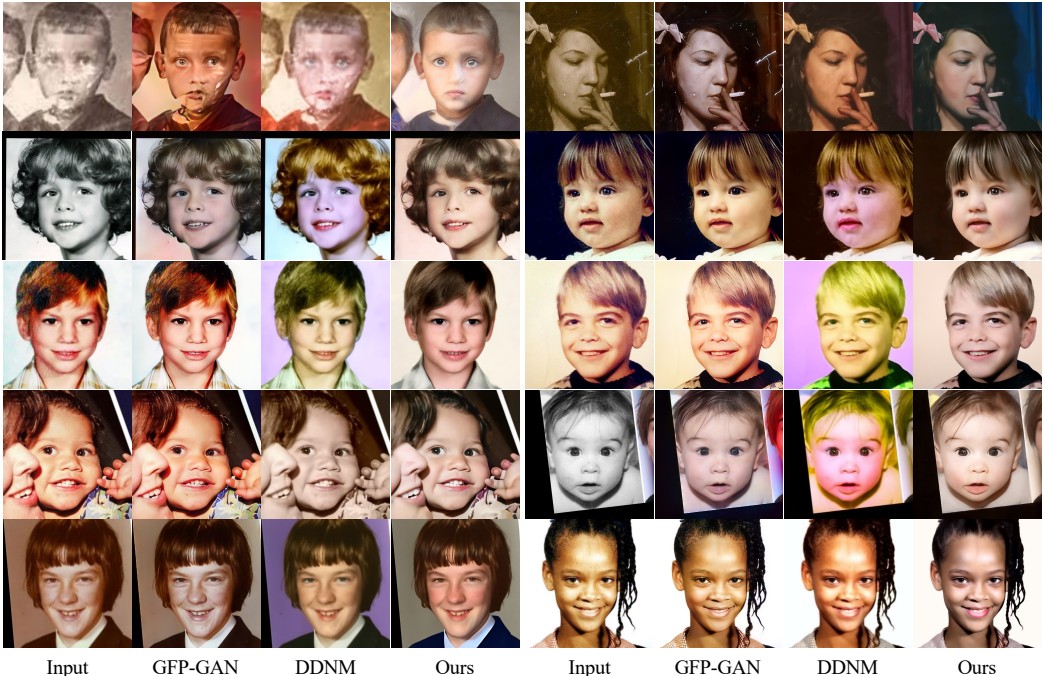

| Input | GFP-GAN | DDNM | Ours | Input | GFP-GAN | DDNM | Ours |

Figure 11: **Comparison on Old Photo Restoration on Challenging Cases.** Our PGDiff is able to produce high-quality restored outputs with natural color and complete faces.

## B.3 More Results on Reference-Based Restoration

We provide more visual results on the reference-based restoration in Fig. 12, which is our exploratory extension based on blind face restoration. Test images come from the CelebRef-HQ dataset [24], which contains $1,005$ entities and each person has $3$ to $21$ high-quality images. With identity loss added, we observe that our method is able to produce personal characteristics similar to those of the ground truth.

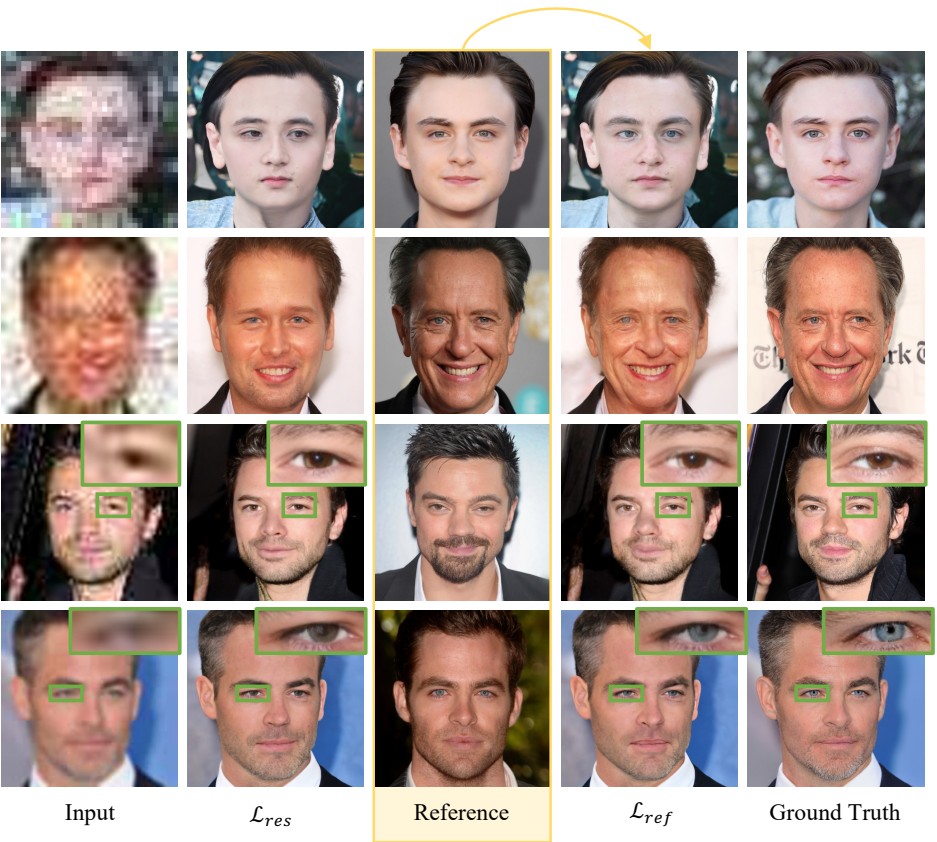

|  |  |  |  |  |
|---|---|---|---|---|
| Input | $\mathcal{L}_{res}$ | Reference | $\mathcal{L}_{ref}$ | Ground Truth |

Figure 12: **Comparison on Reference-Based Face Restoration.** Our PGDiff produces personal characteristics which are hard to recover without reference.

## B.4 More Results on Face Inpainting

In this section, we provide quantitative and more qualitative comparisons with state-of-the-art methods in Fig. 13, including **(1)** task-specific methods: GPEN [44] and CodeFormer [48] and **(2)** diffusion-prior-based methods: GDP [9] and DDNM [42]. As shown in Fig. 4 and Fig. 13, since DDNM and CodeFormer are relatively more competitive than others, we make quantitative comparisons of our methods against them.

We can observe from Fig. 13 that our method is able to recover challenging structures such as glasses. Moreover, diverse and photo-realism outputs can be obtained by setting different random seeds. As for quantitative comparisons in Table 5, we believe that the ability to produce diverse results is also crucial in this task, and the evaluation should not be constrained to the similarity to the ground truth. Thus, we opt to employ NR metrics including FID, NIQE, and MUSIQ instead of FR metrics. Regarding the MUSIQ metric, it has been trained on different datasets featuring various purposes [21]. For inpainting, we employ MUSIQ-KonIQ which focuses on quality assessment. Our method is able to achieve the highest score across all metrics.

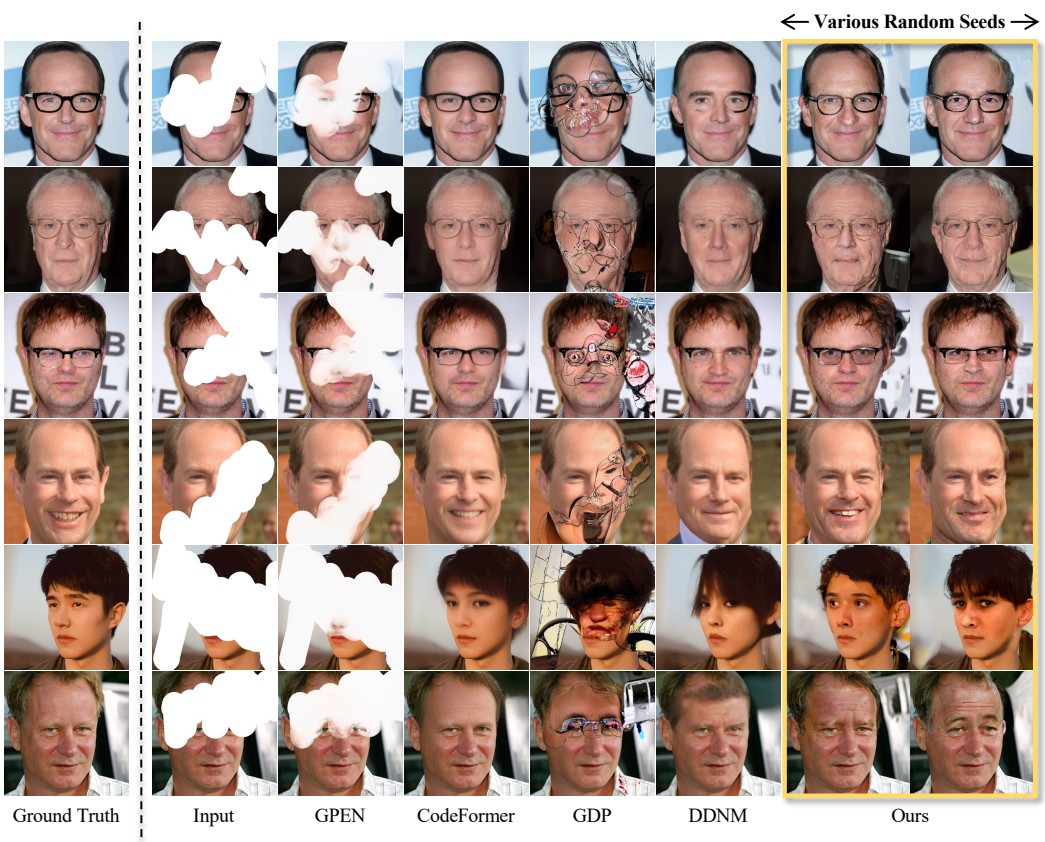

Figure 13: **Comparison on Face Inpainting on Challenging Cases.** Our PGDiff produces natural outputs with pleasant details coherent to the unmasked regions. Moreover, different random seeds give various contents of high quality.

## B.5   More Results on Face Colorization

In this section, we provide quantitative and more qualitative comparisons with state-of-the-art methods in Fig. 14, including **(1)** task-specific methods: CodeFormer [48] and **(2)** diffusion-prior-based methods: GDP [9] and DDNM [42]. As shown in Fig. 3 and Fig. 14, since DDNM and CodeFormer are relatively more competitive than others, we make quantitative comparisons of our methods against them.

We can observe from Fig. 14 that our method produces more vibrant colors and finer details than DDNM and CodeFormer. Moreover, our method demonstrates a desirable diversity by guiding with various color statistics. As for quantitative comparisons in Table 5, we believe that the ability to produce diverse results is also crucial in this task, and the evaluation should not be constrained to the similarity to the ground truth. Thus, we opt to employ NR metrics including FID, NIQE, and MUSIQ instead of FR metrics. Regarding the MUSIQ metric, it has been trained on different datasets featuring various purposes [21]. For colorization, we choose MUSIQ-AVA that puts more emphasis on aesthetic assessment. Although CodeFormer has a better score in NIQE, it clearly alters the input identity (see Fig. 14) and requires training a separate model for each task. On the contrary, our method requires only a pre-trained diffusion model for both inpainting and colorization, and is able to achieve best scores across almost all metrics.

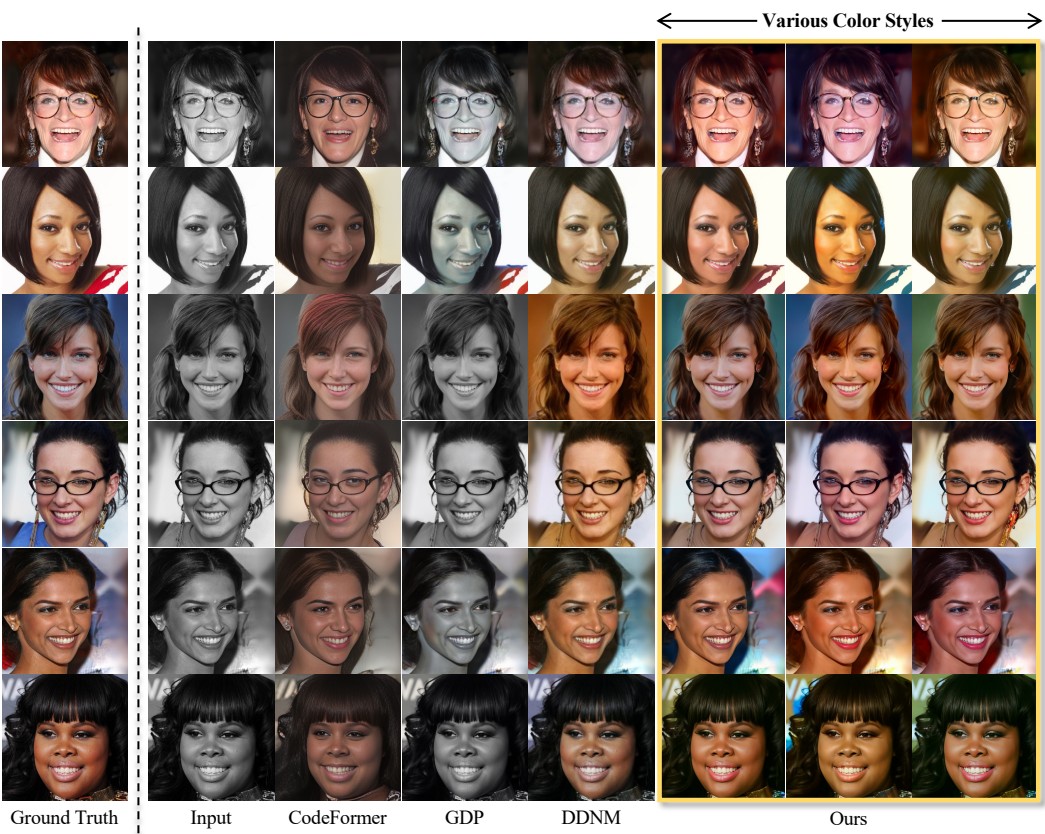

Figure 14: **Comparison on Face Colorization.** Our PGDiff produces diverse colorized outputs with various color statistics given as guidance.

## C   Limitations

As our PGDiff is based on a pre-trained diffusion model, our performance largely depends on the capability of the model in use. In addition, since a face-specific diffusion model is adopted in this work, our method is applicable only to faces in its current form. Nevertheless, this problem can be resolved by adopting stronger models trained for generic objects. For example, as shown in Fig. 15, we employ an unconditional $256 \times 256$ diffusion model trained on the ImageNet dataset [30] provided by [7], and achieve promising results on inpainting and colorization. Further exploration on natural scene restoration will be left as our future work.

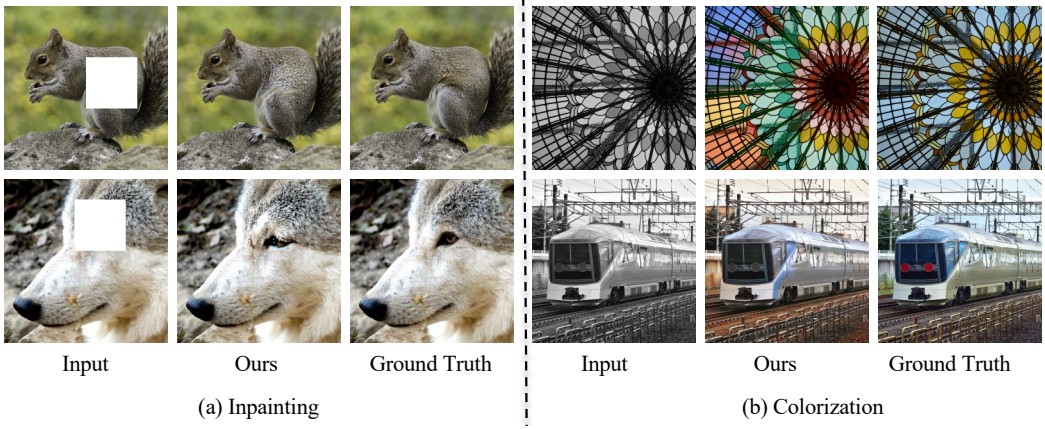

| Input | Ours | Ground Truth | Input | Ours | Ground Truth |

(a) Inpainting   (b) Colorization

Figure 15: Extension on natural images for the inpainting and colorization tasks. By employing an unconditional $256 \times 256$ diffusion model trained on the ImageNet dataset [30] provided by [7], our method achieves promising results.

## D   Broader Impacts

This work focuses on restoring images corrupted by various forms of degradations. On the one hand, our method is capable of enhancing the quality of images and improving user experiences. On the other hand, our method could generate inaccurate outputs, especially when the input is heavily corrupted. This could potentially lead to deceptive information, such as incorrect identity recognition. In addition, similar to other restoration algorithms, our method could be used by malicious users for data falsification. We advise the public to use our method with care.

