[18]. Since the pre-trained diffusion model DDNM employs [10] is trained on the CelebA-HQ dataset [4], we take the CelebRef-HQ [8] dataset for testing. Our method is able to recover challenging structures such as glasses. Moreover, diverse and photo-realism outputs can be obtained by setting different random seeds.

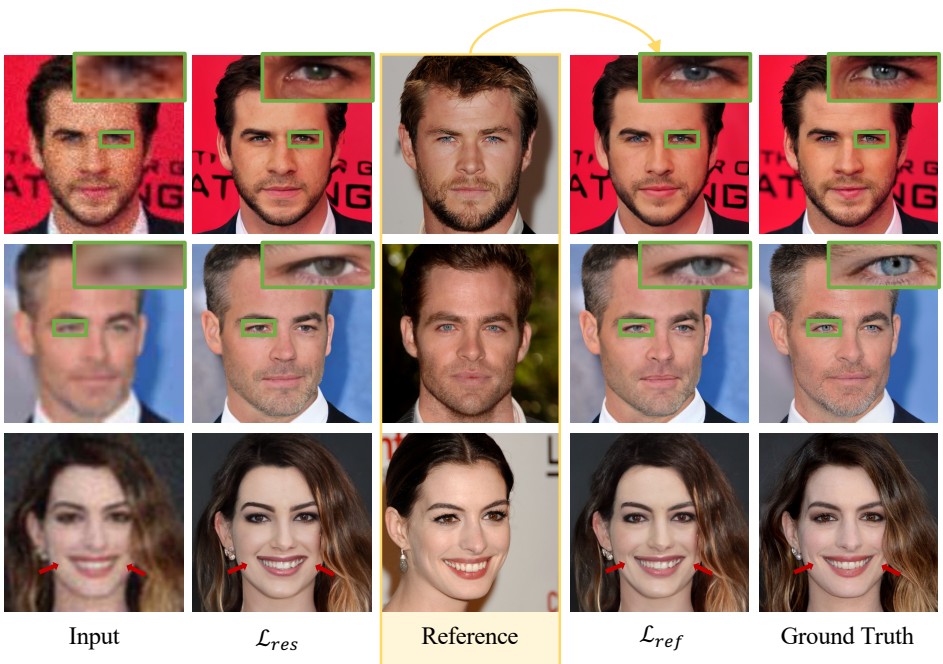

| Input | $\mathcal{L}_{res}$ | Reference | $\mathcal{L}_{ref}$ | Ground Truth |

Figure 4: **Comparison on Reference-Based Face Restoration.** Our method produces personal characteristics which are hard to recover without reference.

← **Various Random Seeds** →

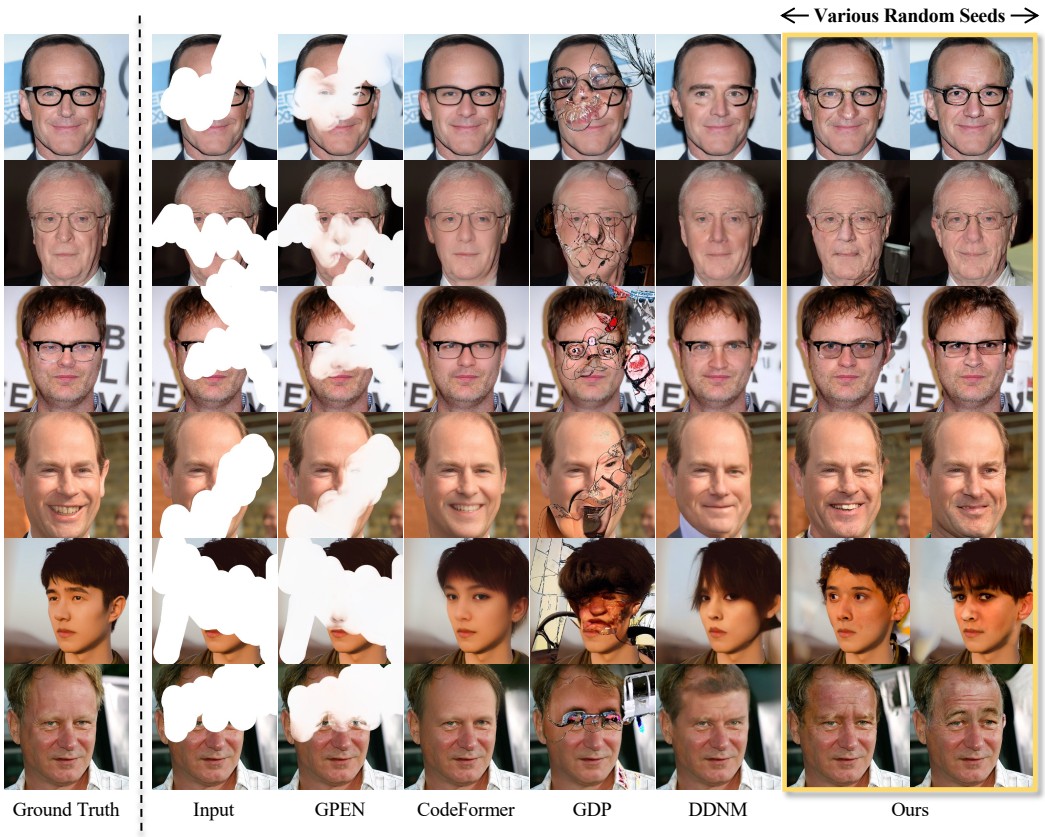

| Ground Truth | Input | GPEN | CodeFormer | GDP | DDNM | Ours |

Figure 5: **Comparison on Face Inpainting on Challenging Cases.** Our method produces natural outputs with pleasant details coherent to the unmasked regions. Moreover, different random seeds give various contents of high quality.

## B.5 More Results on Face Colorization

In this section, we provide more qualitative comparisons with state-of-the-art methods in Fig. 6, including **(1)** task-specific methods: CodeFormer [21] and **(2)** diffusion-prior-based methods: GDP [2] and DDNM [18]. Even though the test images come from the CelebA-HQ dataset [4], our method still produces more vibrant colors and finer details than DDNM. Moreover, our method demonstrates a desirable diversity by guiding with various color statistics.

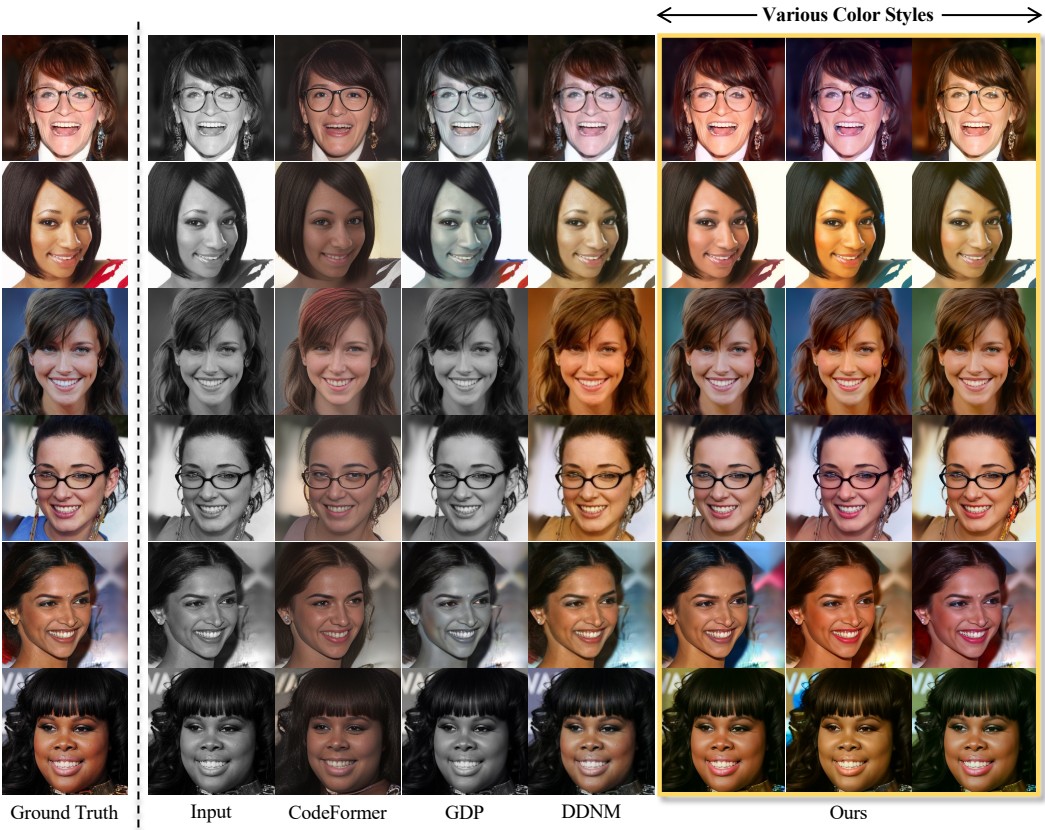

Figure 6: **Comparison on Face Colorization.** Our method produces diverse colorized outputs with various color statistics given as guidance.

## C   Limitations

As our partial guidance is based on a pre-trained diffusion model, our performance largely depends on the capability of the model in use. In addition, since a face-specific diffusion model is adopted in this work, our method is applicable only on faces in its current form. Nevertheless, this problem can be resolved by adopting stronger models trained for generic objects. For example, as shown in Fig. 7, we employ an unconditional $256 \times 256$ diffusion model trained on the ImageNet dataset [13] provided by [1], and achieve promising results on inpainting and colorization. Further exploration on natural scene restoration will be left as our future work.

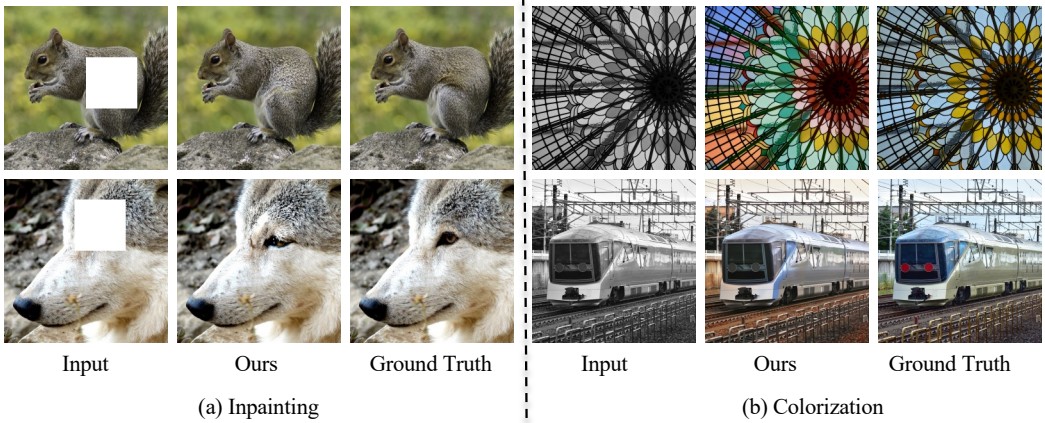

| Input | Ours | Ground Truth | Input | Ours | Ground Truth |

(a) Inpainting (b) Colorization

Figure 7: Extension on natural images for the inpainting and colorization tasks. By employing an unconditional $256 \times 256$ diffusion model trained on the ImageNet dataset [13] provided by [1], our method achieves promising results.

## D   Broader Impacts

This work focuses on restoring images corrupted by various forms of degradations. On the one hand, our method is capable of enhancing the quality of images, improving user experiences. On the other hand, our method could generate inaccurate outputs, especially when the input is heavily corrupted. This could potentially lead to deceptive information, such as incorrect identity recognition. In addition, similar to other restoration algorithms, our method could be used by malicious users for data falsification. We advise the public to use our method with care.