# OpenReview forum: "PGDiff: Guiding Diffusion Models for Versatile Face Restoration via Partial Guidance"
_NeurIPS.cc/2023/Conference — NeurIPS 2023 poster_

### Official Review · Reviewer_ggEw · 2023-06-07

**Soundness:** 3 good
**Presentation:** 3 good
**Contribution:** 3 good
**Rating:** 6
**Confidence:** 4

**Summary:**

The current restoration approaches based on diffusion prior rely on prior knowledge of the degradation process,  and thus fail to seamlessly adapt to different scenarios. Motivated by this, the paper proposed a " partial guidance" approach to directly modeling the distribution properties of high-quality images and then exploit it to serve as guidance for the diffusion process. The extensive experiments demonstrate the advantage of the proposed approach.

**Strengths:**

- The proposed method diverges from traditional practices of modeling the degradation process, and instead, focuses on modeling the desired visual properties of high-quality images. The learned vision cues are then served as diffusion guidance for the generation process. This motivation and the proposed model are technically sound to me. As demonstrated in the paper, it showcases adaptability to different degradation situations, yielding outstanding results.

- The experiments conducted in the study seems solid, featuring comprehensive comparisons with various baseline approaches.

**Weaknesses:**

- For face restoration, it's important to make sure the restoration is performed in an identity-preserving way, and quantitative evaluation regarding this is crucial. However, such quantitative evaluation seems to be missing in the paper.

- To clarify further,  I have no doubt the image quality of the proposed approach outperforms the baseline, but I'm concerned if the quality is improved at the cost of degradation in identity. So it's important to get a quantitative evaluation regarding this

**Questions:**

How would the proposed method compare to the pre-trained text2img diffusion model?

- Let me explain for further clarification: with the current pre-trained text2img diffusion model (stable diffusion, for example), one can perform image enhancement using img2img pipeline,  where you start with an inversed image (plus a small amount of noise) and guide the diffusion process with text prompt like "a high-resolution high-quality photo".  It would be interesting to add such an experiment for comparison.

- for you reference, by "img2img" pipeline I mean: https://huggingface.co/docs/diffusers/api/pipelines/stable_diffusion/img2img

**Limitations:**

No major limitation in terms of societal impact

---

> ### Author Rebuttal · Authors · 2023-08-10
>
> **Identity-preserving evaluation on blind face restoration.** We provide a quantitative evaluation regarding identity-preserving issues in the table below. Considering the importance of identity-preserving in blind face restoration, we introduced reference-based restoration in Sec. 4.5 of the manuscript in addition to the general restoration in Sec. 4.1. Please note that existing methods do not have a mechanism to accept additional identity or reference guidance.
>
> Since we employ heavy degradation settings when synthesizing CelebRef-HQ, it is noteworthy that identity features are largely distorted in severely corrupted input images (see Fig. 6 of the rebuttal material). Thus, it is almost impossible to predict an identity-preserving face without any additional identity information. However, with our reference-based restoration, we observe that the restored face with a high-quality reference image of the same person helps generate personal characteristics which are very similar to the ground truth. The large enhancement of identity preservation can also be seen in the table below, where our reference-based methods achieve the highest IDS, increasing by 0.32 and significantly higher than the other methods. Thanks to the flexibility of our framework, high-quality reference images effectively contribute to restoring low-quality inputs especially in terms of identity preservation.
>
> |   *Blind Restoration*  | **Ours (w/ ref)** | **Ours (w/o ref)** | **DifFace** | **GFPGAN** | **CodeFormer** |
> |-----------|:------------:|:--------:|:-----------:|:----------:|:--------------:|
> | **IDS**$\uparrow$   | 	**0.76** 	|   0.44   | 	0.56*	|	0.36	|  	0.55  	|
>
> **Comparison between our method and text2img diffusion model.** As suggested by the reviewer ggEw, we compare the results generated by img2img stable diffusion model guided by different text prompts in Fig. 5 of the rebuttal material. We observe that a general text prompt like “A high-resolution high-quality photo” fails to generate a truly high-quality restored output given a severely corrupted input. Adding more detailed descriptions (e.g., “boy”, “smiling boy”, and “grinning boy”) of the input image helps enhance the output quality, but at a cost of significant loss of fidelity and identity. By comparison, our method obviously outperforms the text2img diffusion model in terms of both quality and fidelity.

---

> > ### Author Response · Authors · 2023-08-16
> >
> > We appreciate your reviews and comments. Since it is close to the end of the discussion period, may we ask if our rebuttal has resolved your concerns?
> >
> > If you have any further questions, please do not hesitate to reply to our responses. Thank you again for all the constructive comments.

---

> > > ### Comment · Reviewer_ggEw · 2023-08-21
> > >
> > > Thank you for your rebuttal. I believe the rebuttal addressed my concerns and I would recommend acceptance.

---

> > > > ### Author Response · Authors · 2023-08-21
> > > >
> > > > We are pleased that our response has addressed your concerns. Your invaluable comments are deeply appreciated!

---

### Official Review · Reviewer_hvsJ · 2023-06-30

**Soundness:** 3 good
**Presentation:** 3 good
**Contribution:** 2 fair
**Rating:** 3
**Confidence:** 5

**Summary:**

This paper proposes to use some simple properties to guide the reverse diffusion process. The proposed approach makes no assumptions about the degradation process. This paper also shows many different face restoration visual results to demonstrate the superiority of the proposed method.


**Strengths:**

1.	The proposed method has been experimented with in many different face restoration tasks, which demonstrates that the proposed method can be easily adapted to other face restoration tasks.
2.	Overall, this paper is well-writing and easy to understand.


**Weaknesses:**

1.	The main contribution of this paper is using some simple properties to guide the reverse diffusion process, which is inherited from classifier guidance [7]. What is the technical and original contribution of this paper? Please discuss the technical differences between the proposed work and classifier guidance [7].
2.	How to choose the properties that are used in different tasks? This deserves an ablation study. Please show the influence of some different properties.
3.	This paper claims one of the advantages of the proposed method is no assumption of the degradation process. As far as I know, the DifFace also does not need to know the degradation process. What is the advantage of the proposed method against DifFace?
4.	No quantitative results are found in the manuscript. Only visual results are not enough to evaluate the proposed method. Please provide a fair quantitative comparison against other comparing methods such as commonly used datasets CelebA-Test, and three real-world testing datasets proposed in GFPGAN.
5.	The inference time is also important to evaluate the proposed method. Please provide the inference time comparison again with other comparing methods.
6.	Some failure cases can contribute to understanding the limitation of the proposed method. Please show some failure cases.


**Questions:**

Please see Weakness.

**Limitations:**

This paper proposes to use some simple properties to guide the reverse diffusion process. But this paper has not discussed the limitations of the proposed method. I suggest the author discuss the potential and limitations of the proposed method in more detail such as showing some failure cases, which will make the contributions of the proposed method more significant.

---

> ### Author Rebuttal · Authors · 2023-08-10
>
> **Technical differences between our method and classifier guidance.** To model the desired properties of high-quality images, we devise an instantiation named partial guidance by adapting classifier guidance in image restoration (IR) tasks. While the rough idea is inherited from the classifier guidance framework, we extend the classifier from a label predictor to a dense predictor corresponding to a specific image property in the image restoration task. More importantly, we propose special design for the classifier network (see A.3 in the supplementary), dynamic guidance scheme (see Sec. 3.2 of the manuscript), and composite guidance (see Sec. 3.2 of the manuscript), which are all crucial to allow flexible property combinations solving versatile image restoration tasks. To our knowledge, we are the first to extend classifier guidance to an IR paradigm where the degradation process is not needed.
>
> **Choice of Properties.** The choice of properties for each task is listed at Tab. 1 of the manuscript. The selected properties are common ones involved in each task. For example, lightness and color values make up a colorized image. In colorization tasks, it is natural for us to guide on these two properties. For blind face restoration, guidance may only come from the input image. We find that using a clean version of the input image gives better performance than using the original input image (see Fig. 4 of the rebuttal material). As for extending general face restoration to reference-based restoration, identity features (e.g., ArcFace embedding) are widely used to extract personal characteristics. Thus, we propose to guide on the identity features of a reference image of the same identity as the input.
>
> **Advantages of the proposed method against DifFace.** Our method proposes a more general framework than DifFace, and DifFace is in fact a special case of our method when tackling blind face restoration. Specifically, DifFace is equivalent to applying partial guidance at only one iteration step multiple times until convergence. As for the application scope, while DifFace is confined to only blind face restoration, our method is versatile at multiple tasks thanks to the flexibility of our framework. Furthermore, extensive qualitative and quantitative results together show that our method achieves better quality and fidelity in the blind face restoration task than DifFace.
>
> **Quantitative evaluation and user study.** Please refer to the global response in "Author Rebuttal" for detailed elaboration.
>
> **Inference time.** Table below shows the inference time comparison with other methods. The results are tested on GeForce RTX 3090, with $512 \times 512$ input images. Although the submitted version is based on DDPM, which takes 1000 steps during sampling, we also provide an accelerated version by using DDIM without a major performance drop (FID: [DDPM] 115.99 vs [DDIM] 117.99 on inpainting for example). Further acceleration such as distillation is left as our future work. The inference time can thus be about 10 times faster, which is comparable to almost all the other diffusion-based methods.
>
> | *Inference Time* | **Ours (DDPM, 1000 steps)** | **Ours (DDIM, 100 steps)** | **GDP** | **DDNM** | **DifFace** | **CodeFormer** | **GFPGAN** |
> |----------|----------|-----------|---------|----------|-------------|----------------|------------|
> | Time (s/img) | 118.18   |   12.02    	| 149.62  | 15.78	| 3.96    	| 0.05       	| 0.03   	|
>
> **Limitations.** Please refer to the global response in "Author Rebuttal" for detailed elaboration.

---

> > ### Comment · Reviewer_hvsJ · 2023-08-12
> > **Official Comment by Reviewer hvsJ**
> >
> > Thanks for the rebuttal and only some of my concerns are addressed.
> >
> > Given the extensive nature of the revisions required for the manuscript, I maintain my recommendation.

---

> > > ### Author Response · Authors · 2023-08-14
> > >
> > > We understand that you have some remaining concerns, and we apologize for any oversight in fully addressing them all in the rebuttal.
> > >
> > > Please let us know what your remaining concerns are after reading our rebuttal. We will carefully review your comments once again and try our best to address your concerns.

---

> > > ### Author Response · Authors · 2023-08-21
> > >
> > > Dear Reviewer hvsJ,
> > >
> > > As we approach the conclusion of the discussion period, we wish to revisit your outstanding concerns to ensure that we can provide additional clarification or make necessary revisions. If there are any unresolved matters that necessitate further attention, please kindly inform us.
> > >
> > > We sincerely value your valuable time and dedication in assisting us in improving the quality of our manuscript.
> > >
> > > Best Regards,
> > > The Authors

---

### Official Review · Reviewer_FgM3 · 2023-07-05

**Soundness:** 3 good
**Presentation:** 2 fair
**Contribution:** 3 good
**Rating:** 5
**Confidence:** 4

**Summary:**

	This paper proposes partial guidance, an approach exploiting pre-trained diffusion models for face restoration. Instead of making assumption about the specific degradation process, partial guidance models properties of high-quality images such as structure and color statistics to implement classifier-guidance during the reverse diffusion process. This approach suits a range of restoration task and can be extended to composite tasks. Experiments demonstrate the effectiveness of the proposed method.

**Strengths:**

-	This paper considers exploiting image properties irrelevant to specific degradation process to tackle versatile face restoration problems, which has demonstrated effectiveness in experiments.
-	The qualitative results of the proposed method provided in the paper and the supplementary material are sufficient and impressive.


**Weaknesses:**

-	Except blind face restoration task, more quantitative results are expected in the remaining tasks like face colorization and face inpainting.
-	In blind face restoration task, the pretrained restorer (such as Real-ESRGAN stated in the supplementary material) is employed and finetuned to predict smooth semantics, but these intermediate results are missing, and the improvement of the proposed method compared to the pretrained/finetuned restorer is not demonstrated or analyzed.


**Questions:**

-	In blind face restoration task, in addition to real-world data, why synthetic data (like degraded CelebA/CelebA-HQ) is not used for evaluation of more quantitative metrics (like PSNR, SSIM, LPIPS), which is adopted in previous work like DifFace.
-	Other concerns have already been mentioned in Weakness.


**Limitations:**

Yes, the authors have addressed the limitations and potential negative societal impact of their work.

---

> ### Author Rebuttal · Authors · 2023-08-10
>
> **Quantitative evaluation and user study.** We believe that PSNR and SSIM fail to reflect the true image quality. For example, in Tab.1 of CodeFormer, input images achieve the third highest PSNR scores and the highest SSIM scores across all methods. Thus, we exclude them from full-reference metrics in our evaluation and employ LPIPS and IDS instead. Please refer to the global response in "Author Rebuttal" for detailed elaboration.
>
> **Visualization of intermediate results.** As shown in Fig. 3 of the rebuttal material, given an input low-quality image, we visualize intermediate results during the sampling process. $f(y_0,x_t,t)$ shows the smooth semantics predicted by a pretrained restorer at time $t$, which serves as guidance for sampling $x_{t-1}$. We observe that the pretrained restorer can only predict faces without rich details and sharpness at the early stage. Thanks to our design of including $x_t$ alongside $y_0$ as the input to $f$, the prediction results of the pretrained restorer benefit from $\hat{x} _0$ which is growing sharper, and in turn enhance the sharpness of the $x _{t-1}$ sampled.

---

> > ### Author Response · Authors · 2023-08-16
> >
> > We appreciate your reviews and comments. Since it is close to the end of the discussion period, may we ask if our rebuttal has resolved your concerns?
> >
> > If you have any further questions, please do not hesitate to reply to our responses. Thank you again for all the constructive comments.

---

> > ### Comment · Reviewer_FgM3 · 2023-08-18
> >
> > Thank you for your rebuttal.
> > I will increase my score to 5 since most of my concerns are addressed.

---

> > > ### Author Response · Authors · 2023-08-21
> > >
> > > We are pleased that our response has addressed your concerns. Your invaluable comments are deeply appreciated!

---

### Official Review · Reviewer_GVHx · 2023-07-05

**Soundness:** 2 fair
**Presentation:** 3 good
**Contribution:** 3 good
**Rating:** 6
**Confidence:** 5

**Summary:**

This paper proposes a novel solution for blind face restoration. Instead of modeling the degradation process, the authors propose to model the desired properties of high-quality images as classifiers. Similar to guided diffusion, the authors guide the diffusion generation process with specific classifiers to achieve image restoration. Visual results well proved the effectiveness of the proposed methods.

**Strengths:**

1. The idea of modeling the desired properties of high-quality images as classifiers is novel and interesting.
2. The proposed method can solve blind image restoration and can restore images following the reference properties, e.g., color and identity, which is novel and practical.
3. The paper is well-written and easy to follow.

**Weaknesses:**

1. There seems no quantitative evaluation. It is not persuasive enough with only visual comparisons.
2. The setting of hyperparameters may be difficult.
3. Some overclaims. For example, the author said in line 115, "Our partial guidance does not assume any prior knowledge of the degradation process." However, PG needs to know the degradation type. For inpainting, the mask is also needed. (2)

**Questions:**

1. Please see the weaknesses.
2. I am curious about the effectiveness of PG on natural image restoration. For example, what about applying PG on stable diffusion?
3. For the colorization task, the lightness constraint is linear and can be analytically solved in a similar way to DDNM. I wonder if it works to simply apply Color Statistics to DDNM?

**Limitations:**

Though the evaluation is poor, I still give a relatively positive score for the time being and hope that the authors can supplement quantitative experiments (and subjective questionnaires for blind restoration) and objectively discuss the problems existing in practical applications. After all, this work is about blind restoration. If there is a new method that works well, it will be very helpful to the community.

---

> ### Author Rebuttal · Authors · 2023-08-10
>
> **Quantitative evaluation and user study.** Please refer to the global response in "Author Rebuttal" for detailed elaboration.
>
> **Setting of hyperparameters.** As mentioned in A.2 of the supplementary material, while it is principally flexible to tune the hyperparameters case by case, we provided a set of default parameter choices for each homogeneous task in Table 1.
>
> **Statement of our agnosticism to the degradation process.** We intended to emphasize the agnosticism of degradation process on blind restoration. Existing works that employ guided diffusion for image restoration (DDRM, DDNM, GDP) require prior knowledge on the degradation process (e.g., downsampling kernels). By modeling the desired properties of high-quality images, we bypass this requirement. Thank you for pointing out the inpainting case where a mask is needed, and we will revise our claim to exclude the inpainting task.
>
> **Applying partial guidance on natural image restoration.** Due to the different nature of human faces and natural images, we plan to explore partial guidance on natural images in our future work. Nevertheless, we have made some early explorations demonstrating the feasibility. We provide some natural image restoration results in Fig. 1 of the rebuttal material. In addition, we also provided natural image inpainting and colorization results in Fig. 7 of the supplementary material. Since stable diffusion performs diffusion process in the latent space, it requires careful design of appropriate properties in cases where existing ones are not applicable. This aspect will be further investigated in our future work.
>
> **Applying color statistics on DDNM.** As shown in Fig. 2 of the rebuttal material, we apply our partial guidance on DDNM by guiding on $\hat{x} _{0|t}$ (computed by DDNM) with AdaIN($\hat{x} _{0|t}$). DDNM produces more vivid and natural colorized outputs with various color statistics given as guidance. It shows that our partial guidance can be flexibly applied in the DDNM framework for further improvement.
>
> **Limitations.** Please refer to the global response in "Author Rebuttal" for detailed elaboration.

---

> > ### Comment · Reviewer_GVHx · 2023-08-12
> >
> > Your rebuttal dispelled most of my concerns and I will increase my score to 6.

---

> > > ### Author Response · Authors · 2023-08-12
> > >
> > > We are glad that our response can resolve your concerns. Thank you so much for your invaluable comments!

---

### Author Rebuttal · Authors · 2023-08-10

We are encouraged that the reviewers find our work novel and interesting [Reviewer GVHx, ggEw]; practical and versatile in multiple image restoration tasks [Reviewer GVHx, FgM3, hvsJ, ggEw]; presenting impressive and outstanding visual results [Reviewer GVHx, FgM3, ggEw]; well-written and easy to follow [Reviewer GVHx, hvsJ].

In light of the common concerns about more quantitative evaluation and discussion on limitations of our method, we supplement with a detailed discourse on these matters here.

**Quantitative evaluation and user study.**

**For blind face restoration.** We provided a quantitative comparison in Tab. 2 of the supplementary, evaluating on three *real-world* datasets LFW-Test, WebPhoto-Test, and WIDER-Test. Our method achieves best or second-best scores across all three datasets for both FID and NIQE metrics.

We also include a quantitative evaluation on the *synthetic* CelebRef-HQ dataset in the table below. Considering the importance of identity-preserving in blind face restoration, we introduced reference-based restoration in Sec. 4.5 of the manuscript in addition to the general restoration in Sec. 4.1. As shown in Fig. 2 of both the manuscript and supplementary materials, since DifFace, GFPGAN, and CodeFormer are relatively more competitive than others, we make quantitative comparisons of our methods against them. The table below shows that our methods achieve best or second best scores across both no-reference (NR) metrics for image quality (i.e., FID and MUSIQ) and full-reference (FR) metrics for identity preservation (i.e., LPIPS and IDS).

Since we employ heavy degradation settings when synthesizing CelebRef-HQ, it is noteworthy that identity features are largely distorted in severely corrupted input images (see Fig. 6 of the rebuttal material). Thus, it is almost impossible to predict an identity-preserving face without any additional identity information. Nevertheless, with our reference-based restoration, we observe that the restored face with a high-quality reference image of the same person helps generate personal characteristics that are highly similar to the ground truth. The large enhancement of identity preservation is also indicated in the table below, where our reference-based method achieves the highest IDS, increasing by 0.32.

|   *Blind Restoration*  | **Ours (w/ ref)** | **Ours (w/o ref)** | **DifFace** | **GFPGAN** | **CodeFormer** |
|-----------|:------------:|:--------:|:-----------:|:----------:|:--------------:|
| **FID**$\downarrow$   |	121.25*	|  **119.98**  |	123.18   |   186.88   | 	129.17 	|
| **MUSIQ-KonIQ**$\uparrow$ | 	64.67	|   67.26*  |	60.98	|	63.33   |	  **69.62** 	|
| **LPIPS**$\downarrow$ | 	0.35* 	|   **0.34**   | 	0.35*	|	0.49	|  	0.36  	|
| **IDS**$\uparrow$   | 	**0.76** 	|   0.44   | 	0.56*	|	0.36	|  	0.55  	|

In addition to the qualitative and quantitative evaluations on blind face restoration, the results of a user study are shown in the table below, with an involvement of 30 participants. Each participant is shown 30 randomly sampled image triplets and asked to select a visually better restored output. Each triplet is composed of one corrupted input, one from our method, and one from a randomly chosen method. It is observed that our results are preferred by participants compared to DifFace and GFPGAN. Considering CodeFormer is a task-specific method that requires careful multi-stage training, our method is able to achieve a comparable performance with CodeFormer even without extensive training on the task.

|           *User Study*      | **DifFace** | **GFPGAN** | **CodeFormer** |
|-----------------|:--------:|:-------:|:--------------:|
| **Favoring Ours**$\uparrow$ |  72.60%  |  72.01% |     51.22%     |

**For face inpainting and colorization.** We provide a quantitative evaluation on the *synthetic* CelebA-Test dataset in the table below. We believe that the ability to produce diverse results is also crucial in these two tasks, and the evaluation should not be constrained to the similarity to the ground truth. Thus, we opt to employ NR metrics including FID, NIQE, and MUSIQ instead of FR metrics. Regarding the MUSIQ metric, it has been trained on different datasets featuring various purposes. For inpainting, we employ MUSIQ-KonIQ which focuses on quality assessment, while for colorization, we choose MUSIQ-AVA that puts more emphasis on aesthetic assessment. Although CodeFormer has a better score in NIQE for the colorization task, it clearly alters the input identity (see Fig. 6 of the supplementary) and requires training a separate model for each task. On the contrary, our method requires only a pre-trained diffusion model for both inpainting and colorization, and is able to achieve best scores across almost all metrics. The qualitative results in Fig. 3, 4 of the manuscript and Fig. 5, 6 of the supplementary also prove the superiority of our approach.

| *Inpainting*  	| **Ours** | **CodeFormer** | **DDNM** |
|-----------------|----------|----------------|----------|
| **FID**$\downarrow$     	|  **115.99**  | 	120.93 	|  137.57  |
| **NIQE**$\downarrow$    	|   **3.65**   |  	4.22  	|   5.35   |
| **MUSIQ-KonIQ**$\uparrow$ |   **73.20**  |  	72.48 	|   59.38  |

| *Colorization* | **Ours** | **CF** | **DDNM** |
|------------------|----------|--------|----------|
| **FID**$\downarrow$      	|  **119.31**  | 126.91 |  146.66  |
| **NIQE**$\downarrow$     	|   4.71   |  **4.43**  |   5.11   |
| **MUSIQ-AVA**$\uparrow$|   **5.23**   |  4.91  |   4.07   |

**Limitations.**

As our partial guidance is based on a pre-trained diffusion model, our performance largely depends on the capability of the model in use. The pre-trained face diffusion model we employed is trained on the FFHQ dataset, where side faces seldom exist. As a result, failures tend to occur on side faces (see Fig. 7 in the rebuttal material). Since CodeFormer and DifFace are also trained on the FFHQ dataset, we share similar issues on side faces.

---

### Decision · Program_Chairs · 2023-09-21

**Decision:**

Accept (poster)

**Comment:**

The paper presents a method for face restoration using a mechanism called partial guidance, wherein intermediate outputs of the diffusion model are supervised by a classifier to promote, e.g., consistency with the grayscale input for colorization or identity similarity. Initially the paper received borderline reviews, with reviewers raising concerns about the lack of quantitative evaluation and need for additional comparisons to baseline methods, especially DifFace. The authors responded with a quantitative evaluation of the method compared to baselines for blind restoration, inpainting, and colorization. Three of the four reviewers upgraded their scores, and, after considering the rebuttal and discussion, the AC finds the evaluation and quality of results sufficient to merit acceptance. The authors should include the quantitative evaluation and the additional results requested by reviewers in the camera ready version.